# Open-Ended Instructable Embodied Agents with Memory-Augmented Large Language Models

**Gabriel Sarch    Yue Wu    Michael J. Tarr    Katerina Fragkiadaki**
Carnegie Mellon University
{gsarch,ywu5,mt01}@andrew.cmu.edu,katef@cs.cmu.edu
helper-agent-llm.github.io

## Abstract

Pre-trained and frozen LLMs can effectively map simple scene re-arrangement instructions to programs over a robot's visuomotor functions through appropriate few-shot example prompting. To parse open-domain natural language and adapt to a user's idiosyncratic procedures, not known during prompt engineering time, fixed prompts fall short. In this paper, we introduce HELPER, an embodied agent equipped with as external memory of language-program pairs that parses free-form human-robot dialogue into action programs through retrieval-augmented LLM prompting: relevant memories are retrieved based on the current dialogue, instruction, correction or VLM description, and used as in-context prompt examples for LLM querying. The memory is expanded during deployment to include pairs of user's language and action plans, to assist future inferences and personalize them to the user's language and routines. HELPER sets a new state-of-the-art in the TEACh benchmark in both Execution from Dialog History (EDH) and Trajectory from Dialogue (TfD), with 1.7x improvement over the previous SOTA for TfD. Our models, code and video results can be found in our project's website: helper-agent-llm.github.io.

## 1 Introduction

Parsing free-form human instructions and human-robot dialogue into task plans that a robot can execute is challenging due to the open-endedness of environments and procedures to accomplish, and to the diversity and complexity of language humans use to communicate their desires. Human language often contains long-term references, questions, errors, omissions, or references and descriptions of routines specific to a particular user (Tellex et al., 2011; Liang, 2016; Klein and Manning, 2003). Instructions need to be interpreted in the environmental context in which they are issued, and plans need to adapt in a closed-loop to execution failures.

Large Language Models (LLMs) trained on Internet-scale text can parse language instructions to task plans with appropriate plan-like or code-like prompts, without any finetuning of the language model, as shown in recent works (Ahn et al., 2022; Liang et al., 2022; Zeng et al., 2022; Huang et al., 2022b; Singh et al., 2022a). The state of the environment is provided as a list of objects and their spatial coordinates, or as a free-form text description from a vision-language model (Liang et al., 2022; Liu et al., 2023b; Wu et al., 2023a; Ahn et al., 2022). Using LLMs for task planning requires engineering a prompt that includes a description of the task for the LLM to perform, a robot API with function documentation and expressive function names, environment and task instruction inputs, and a set of in-context examples for inputs and outputs for the task (Liang et al., 2022). These methods are not trained in the domain of interest; rather they are prompt-engineered having in mind the domain at hand.

How can we extend LLM-prompting for semantic parsing and task planning to open-domain, free-form instructions, corrections, human-robot dialogue, and users' idiosyncratic routines, not known at prompt engineering time? The prompts used for the domain of tabletop rearrangement are already approaching the maximum context window of widely used LLMs (Singh et al., 2022a; Liang et al., 2022). Even as context window size grows, more prompt examples result in larger attention operations and cause an increase in both inference time and resource usage.

To this end, we introduce HELPER (Human-instructable Embodied Language Parsing via Evolving Routines), a model that uses retrieval-augmented situated prompting of LLMs to parse free-form dialogue, instructions and corrections from humans and vision-language models to programs over a set of parameterized visuomotor rou-

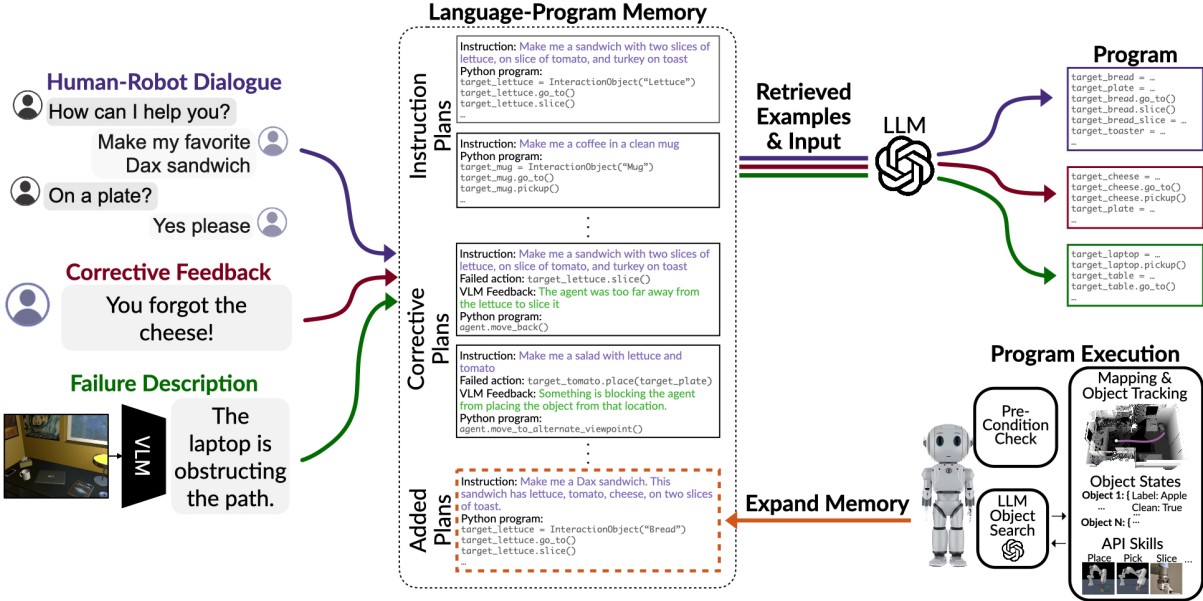

Figure 1: **Open-ended instructable agents with retrieval-augmented LLMs.** We equip LLMs with an external memory of language and program pairs to retrieve in-context examples for prompts during LLM querying for task plans. Our model takes as input instructions, dialogue segments, corrections and VLM environment descriptions, retrieves relevant memories to use as in-context examples, and prompts LLMs to predict task plans and plan adjustments. Our agent executes the predicted plans from visual input using occupancy and semantic map building, 3D object detection and state tracking, and active exploration using guidance from LLMs' common sense to locate objects not present in the maps. Successful programs are added to the memory paired with their language context, allowing for personalized subsequent interactions.

tines. HELPER is equipped with an external non-parametric key-value memory of language - program pairs. HELPER uses its memory to retrieve relevant in-context language and action program examples, and generates prompts tailored to the current language input. HELPER expands its memory with successful executions of user specific procedures; it then recalls them and adapts them in future interactions with the user. HELPER uses pre-trained vision-language models (VLMs) to diagnose plan failures in language format, and uses these to retrieve similar failure cases with solutions from its memory to seed the prompt. To execute a program predicted by the LLM, HELPER combines successful practices of previous home embodied agents, such as semantic and occupancy map building (Chaplot et al., 2020a; Blukis et al., 2022; Min et al., 2021), LLM-based common sense object search (Inoue and Ohashi, 2022), object detection and tracking with off-the-shelf detectors (Chaplot et al., 2020b), object attribute detection with VLMs (Zhang et al., 2022), and verification of action preconditions during execution.

We test HELPER on the TEACh benchmark (Padmakumar et al., 2021), which evaluates agents in their ability to complete a variety of long-horizon household tasks from RGB input given natural language dialogue between a commander (the instruction-giving user) and a follower (the instruction-seeking user). We achieve a new state-of-the-art in the TEACh Execution from Dialog History and Trajectory-from-Dialogue settings, improving task success by 1.7x and goal-condition success by 2.1x compared to prior work in TfD. By further soliciting and incorporating user feedback, HELPER attains an additional 1.3x boost in task success. Our work is inspired by works in the language domain (Perez et al., 2021; Schick and Schütze, 2020; Gao et al., 2020; Liu et al., 2021) that retrieve in-context prompt examples based on the input language query for NLP tasks. HELPER extends this capability to the domain of instructable embodied agents, and demonstrates the potential of memory-augmented LLMs for semantic parsing of open-ended free-form instructive language into an expandable library of programs.

## 2 Related Work

**Instructable Embodied Agents** Significant strides have been made by training large neural

networks to jointly map instructions and their sensory contexts to agent actions or macro-actions using imitation learning (Anderson et al., 2018b; Ku et al., 2020; Anderson et al., 2018a; Savva et al., 2019; Gervet et al., 2022; Shridhar et al., 2020; Cao et al.; Suglia et al., 2021; Fan et al., 2018; Yu et al., 2020; Brohan et al., 2022; Stone et al., 2023; Yu et al., 2023). Existing approaches differ—among others—in the way the state of the environment is communicated to the model. Many methods map RGB image tokens and language inputs directly to actions or macro-actions (Pashevich et al., 2021; Wijmans et al., 2020; Suglia et al., 2021; Krantz et al., 2020). Other methods map language instructions and linguistic descriptions of the environment's state in terms of object lists or objects spatial coordinates to macro-actions, foregoing visual feature description of the scene, in a attempt to generalize better (Liang et al., 2022; Singh et al., 2022a; Chaplot et al., 2020a; Min et al., 2021; Liu et al., 2022a; Murray and Cakmak, 2022; Liu et al., 2022b; Inoue and Ohashi, 2022; Song et al., 2022; Zheng et al., 2022; Zhang et al., 2022; Huang et al., 2022b, 2023; Ahn et al., 2022; Zeng et al., 2022; Huang et al., 2022a). Some of these methods fine-tune language models to map language input to macro-actions, while others prompt frozen LLMs to predict action programs, relying on the emergent in-context learning property of LLMs to emulate novel tasks at test time. Some methods use natural language as the output format of the LLM (Wu et al., 2023a; Song et al., 2022; Blukis et al., 2022; Huang et al., 2022b), and others use code format (Singh et al., 2022a; Liang et al., 2022; Huang et al., 2023). HELPER prompts frozen LLMs to predict Python programs over visuo-motor functions for parsing dialogue, instructions and corrective human feedback.

The work closest to HELPER is LLM Planner (Song et al., 2022) which uses memory-augmented prompting of pretrained LLMs for instruction following. However, it differs from HELPER in several areas such as plan memory expansion, VLM-guided correction, and usage of LLMs for object search. Furthermore, while Singh et al. (2022b) frequently seeks human feedback, HELPER requests feedback only post full task execution and employs Visual-Language Models (VLMs) for error feedback, reducing user interruptions.

Numerous simulation environments exist for

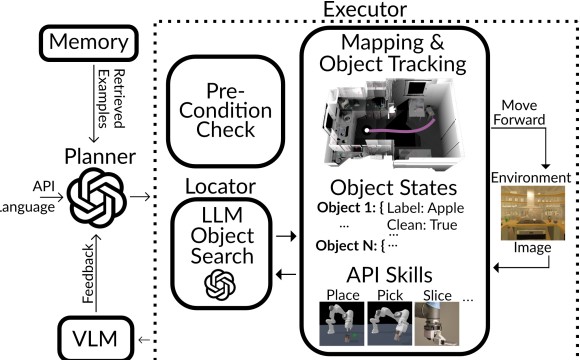

Figure 2: **HELPER's architecture.** The model uses memory-augmented LLM prompting for task planning from instructions, corrections and human-robot dialogue and for re-planning during failures given feedback from a VLM model. The generated program is executed the EXECUTOR module. The EXECUTOR builds semantic, occupancy and 3D object maps, tracks object states, verifies action preconditions, and queries LLMs for search locations for objects missing from the maps, using the LOCATOR module.

evaluating home assistant frameworks, including Habitat (Savva et al., 2019), GibsonWorld (Shen et al., 2021), ThreeDWorld (Gan et al., 2022), and AI2THOR (Kolve et al., 2017). ALFRED (Shridhar et al., 2020) and TEACh (Padmakumar et al., 2021) are benchmarks in the AI2THOR environment (Kolve et al., 2017), measuring agents' competence in household tasks through natural language. Our research focuses on the 'Trajectory from Dialogue' (TfD) evaluation in TEACh, mirroring ALFRED but with greater task and input complexity.

**Prompting LLMs for action prediction and visual reasoning** Since the introduction of few-shot prompting by (Brown et al., 2020), several approaches have improved the prompting ability of LLMs by automatically learning prompts (Lester et al., 2021), chain of thought prompting (Nye et al.; Gao et al., 2022; Wei et al., 2022; Wang et al., 2022; Chen et al., 2022; Yao et al., 2023) and retrieval-augmented LLM prompting (Nakano et al., 2021; Shi et al., 2023; Jiang et al., 2023) for language modeling, question answering, and long-form, multi-hop text generation. HELPER uses memory-augmented prompting by retrieving and integrating similar task plans into the prompt to facilitate language parsing to programs.

LLMs have been used as policies in Minecraft to predict actions (Wang et al., 2023b,a), error correction (Liu et al., 2023b), and for understand-

ing instruction manuals for game play in some Atari games (Wu et al., 2023b). They have also significantly improved text-based agents in text-based simulated worlds (Yao et al., 2022; Shinn et al., 2023; Wu et al., 2023c; Richards, 2023). ViperGPT (Surís et al., 2023), and CodeVQA (Subramanian et al., 2023) use LLM prompting to decompose referential expressions and questions to programs over simpler visual routines. Our work uses LLMs for planning from free-form dialogue and user corrective feedback for home task completion, a domain not addressed in previous works.

## 3 Method

HELPER is an embodied agent designed to map human-robot dialogue, corrections and VLM descriptions to actions programs over a fixed API of parameterized navigation and manipulation primitives. Its architecture is outlined in Figure 2. At its heart, it generates plans and plan adjustments by querying LLMs using retrieval of relevant language-program pairs to include as in-context examples in the LLM prompt. The generated programs are then sent to the EXECUTOR module, which translates each program step into specific navigation and manipulation action. Before executing each step in the program, the EXECUTOR verifies if the necessary preconditions for an action, such as the robot already holding an object, are met. If not, the plan is adjusted according to the current environmental and agent state. Should a step involve an undetected object, the EXECUTOR calls on the LOCATOR module to efficiently search for the required object by utilizing previous user instructions and LLMs' common sense knowledge. If any action fails during execution, a VLM predicts the reason for this failure from pixel input and feeds this into the PLANNER for generating plan adjustments.

### 3.1 PLANNER: Retrieval-Augmented LLM Planning

Given an input $I$ consisting of a dialogue segment, instruction, or correction, HELPER uses memory-augmented prompting of frozen LLMs to map the input into an executable Python program over a parametrized set of manipulation and navigation primitives $G \in \{G_{manipulation} \cup G_{navigation}\}$ that the EXECUTOR can perform (e.g., goto(X), pickup(X), slice(X), ...). Our action API can be found in Section D of the Appendix.

HELPER maintains a key-value memory of language - program pairs, as shown in Figure 3A. Each language key is mapped to a 1D vector using an LLM's frozen language encoder. Given current context $I$, the model retrieves the top-$K$ keys, i.e., the keys with the smallest $L_2$ distance with the embedding of the input context $I$, and adds the corresponding language - program pairs to the LLM prompt as in-context examples for parsing the current input $I$.

Figure 3B illustrates the prompt format for the PLANNER. It includes the API specifying the primitives $G$ parameterized as Python functions, the retrieved examples, and the language input $I$. The LLM is tasked to generate a Python program over parameterized primitives $G$. Examples of our prompts and LLM responses can be found in Section F of the Appendix.

#### 3.1.1 Memory Expansion

The key-value memory of HELPER can be continually expanded with successful executions of instructions to adapt to a user's specific routines, as shown in Figure 1. An additional key-value pair is added with the language instruction paired with the execution plan if the user indicates the task was successful. Then, HELPER can recall this plan and adapt it in subsequent interactions with the user. For example, if a user instructs HELPER one day to *"Perform the Mary cleaning. This involves cleaning two plates and two cups in the sink"*, the user need only say *"Do the Mary cleaning"* in future interactions, and HELPER will retrieve the previous plan, include it in the examples section of the prompt, and query the LLM to adapt it accordingly. The personalization capabilities of HELPER are evaluated in Section 4.4.

#### 3.1.2 Incorporating user feedback

A user's feedback can improve a robot's performance, but requesting feedback frequently can deteriorate the overall user experience. Thus, we enable HELPER to elicit user feedback only when it has completed execution of the program. Specifically, it asks *"Is the task completed to your satisfaction? Did I miss anything?"* once it believes it has completed the task. The user responds either that the task has been completed (at which point HELPER stops acting) or points out problems and corrections in free-form natural language, such as, *"You failed to cook a slice of potato. The potato slice needs to be cooked."*. HELPER uses the language

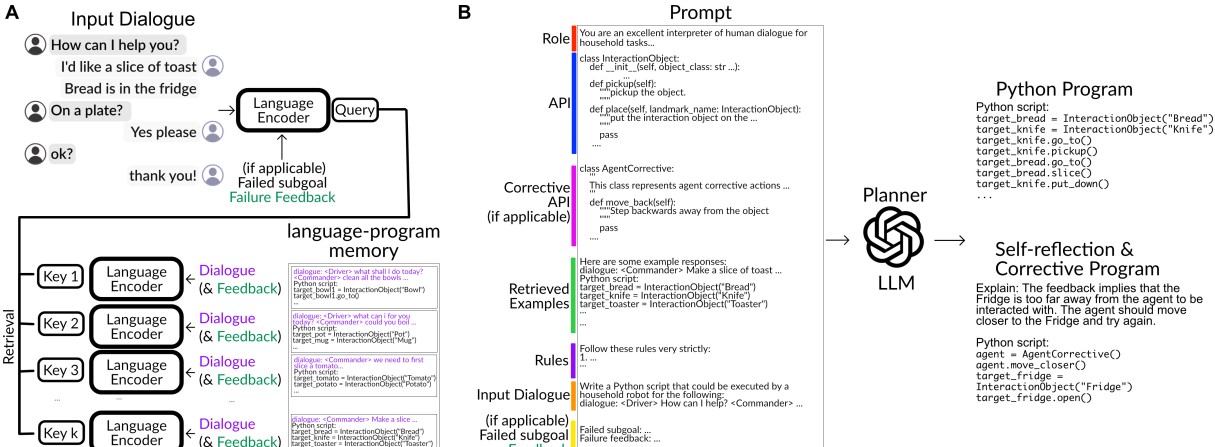

Figure 3: HELPER parses dialogue segments, instructions, and corrections into visuomotor programs using retrieval-augmented LLM prompting. **A.** Illustration of the encoding and memory retrieval process. **B.** Prompt format and output of the PLANNER.

feedback to re-plan using the PLANNER. We evaluate HELPER in its ability to seek and utilize user feedback in Section 4.3.

### 3.1.3 Visually-Grounded Plan Correction using Vision-Language Models

Generated programs may fail for various reasons, such as when a step in the plan is missed or an object-of-interest is occluded. When the program fails, HELPER uses a vision-language model (VLM) pre-trained on web-scale data, specifically the ALIGN model (Jia et al., 2021), to match the current visual observation with a pre-defined list of textual failure cases, such as *an object is blocking you from interacting with the selected object*, as illustrated in Figure 4. The best match is taken to be the failure feedback $F$. The PLANNER module then retrieves the top-$K$ most relevant error correction examples, each containing input dialogue, failure feedback, and the corresponding corrective program, from memory based on encodings of input $I$ and failure feedback $F$ from the VLM. The LLM is prompted with the the failed program step, the predicted failure description $F$ from the VLM, the in-context examples, and the original dialogue segment $I$. The LLM outputs a self-reflection as to why the failure occurred, and generates a program over manipulation and navigation primitives $G$, and an additional set of corrective primitives $G_{corrective}$ (e.g., step-back(), move-to-an-alternate-viewpoint(), ...). This program is sent to the EXECUTOR for execution.

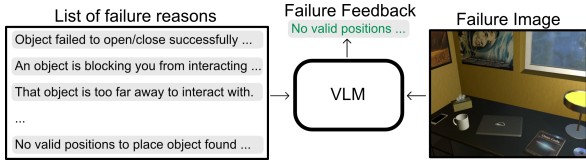

Figure 4: Inference of a failure feedback description by matching potential failure language descriptions with the current image using a vision-language model (VLM).

### 3.2 EXECUTOR: Scene Perception, Pre-Condition Checks, Object Search and Action Execution

The EXECUTOR module executes the predicted Python programs in the environment, converting the code into low-level manipulation and navigation actions, as shown in Figure 2. At each time step, the EXECUTOR receives an RGB image and obtains an estimated depth map via monocular depth estimation (Bhat et al., 2023) and object masks via an off-the-shelf object detector (Dong et al., 2021).

#### 3.2.1 Scene and object state perception

Using the depth maps, object masks, and approximate egomotion of the agent at each time step, the EXECUTOR maintains a 3D occupancy map and object memory of the home environment to navigate around obstacles and keep track of previously seen objects, similar to previous works (Sarch et al., 2022). Objects are detected in every frame and are merged into object instances based on closeness of the predicted 3D centroids. Each object instance is initialized with a set of object state at-

tributes (cooked, sliced, dirty, ...) by matching the object crop against each attribute with the pre-trained ALIGN model (Jia et al., 2021). Object attribute states are updated when an object is acted upon via a manipulation action.

### 3.2.2 Manipulation and navigation pre-condition checks

The EXECUTOR module verifies the pre-conditions of an action before the action is taken to ensure the action is likely to succeed. In our case, these constraints are predefined for each action (for example, the agent must first be holding a knife to slice an object). If any pre-conditions are not satisfied, the EXECUTOR adjusts the plan accordingly. In more open-ended action interfaces, an LLM's common sense knowledge can be used to infer the pre-conditions for an action, rather than pre-defining them.

### 3.2.3 LOCATOR: LLM-based common sense object search

When HELPER needs to find an object that has not been detected before, it calls on the LOCATOR module. The LOCATOR prompts an LLM to suggest potential object search location for the EXECUTOR to search nearby, e.g. "search near the sink" or "search in the cupboard". The LOCATOR prompt takes in the language $I$ (which may reference the object location, e.g., "take the mug from the cupboard" ) and queries the LLM to generate proposed locations by essentially parsing the instruction as well as using its common sense. Based on these predictions, HELPER will go to the suggested locations if they exist in the semantic map (e.g., to the sink) and search for the object-of-interest. The LOCATOR's prompt can be found in Section D of the Appendix.

**Implementation details.** We use OpenAI's `gpt-4-0613` (gpt, 2023) API, except when mentioned otherwise. We resort to the `text-embedding-ada-002` (ada, 2022) API to obtain text embeddings. Furthermore, we use the SOLQ object detector (Dong et al., 2021), which is pretrained on MSCOCO (Lin et al., 2014) and fine-tuned on the training rooms of TEACh. For monocular depth estimation, we use the ZoeDepth network (Bhat et al., 2023), pretrained on the NYU indoor dataset (Nathan Silberman and Fergus, 2012) and subsequently fine-tuned on the training rooms of TEACh. In the TEACh evaluations, we use $K{=}3$ for retrieval.

## 4 Experiments

We test HELPER in the TEACh benchmark (Padmakumar et al., 2021). Our experiments aim to answer the following questions:

1. How does HELPER compare to the SOTA on task planning and execution from free-form dialogue?

2. How much do different components of HELPER contribute to performance?

3. How much does eliciting human feedback help task completion?

4. How effectively does HELPER adapt to a user's specific procedures?

### 4.1 Evaluation on the TEACh dataset

**Dataset** The dataset comprises over 3,000 human–human, interactive dialogues, geared towards completing household tasks within the AI2-THOR simulation environment (Kolve et al., 2017). We evaluate on the Trajectory from Dialogue (TfD) evaluation variant, where the agent is given a dialogue segment at the start of the episode. The model is then tasked to infer the sequence of actions to execute based on the user's intents in the dialogue segment, ranging from MAKE COFFEE to PREPARE BREAKFAST. We show examples of such dialogues in Figures 1 & 3. We also test on the Execution from Dialogue History (EDH) task in TEACh, where the TfD episodes are partitioned into "sessions". The agent is spawned at the start of one of the sessions and must predict the actions to reach the next session given the dialogue and action history of the previous sessions. The dataset is split into training and validation sets. The validation set is divided into 'seen' and 'unseen' rooms based on their presence in the training set. Validation 'seen' has the same room instances but different object locations and initial states than any episodes in the training set. At each time step, the agent obtains an egocentric RGB image and must choose an action from a specified set to transition to the next step, such as pickup(X), turn left(), etc. Please see Appendix Section G for more details on the simulation environment.

**Evaluation metrics** Following evaluation practises for the TEACh benchmark, we use the following two metrics: **1. Task success rate (SR),** which refers to the fraction of task sessions in which the

| | No In-Domain LLM | Memory-Augmented LLM | User Personal-ization | Accepts User Feedback | VLM-Guided correction | LLM-Guided Search | Pre-Condition Check |
|---|---|---|---|---|---|---|---|
| E.T.[Pashevich et al., 2021] | x | x | x | x | x | x | x |
| JARVIS[Zheng et al., 2022] | x | x | x | x | x | x | x |
| FILM[Min et al., 2021, 2022] | x | x | x | x | x | x | x |
| DANLI[Zhang et al., 2022] | x | x | x | x | x | x | ✓ |
| LLM-Planner[Song et al., 2022] | ✓ | ✓ | x | x | x | x | x |
| Code as Policies[Liang et al., 2022] | ✓ | x | x | x | x | x | x |
| HELPER (ours) | ✓ | ✓ | ✓ | ✓ | ✓ | ✓ | ✓ |

Table 1: Comparison of HELPER to previous work.

agent successfully fulfills all goal conditions. **2. Goal condition success rate (GC),** which quantifies the proportion of achieved goal conditions across all sessions. Both of these metrics have corresponding path length weighted (PLW) variants. In these versions, the agent incurs penalties for executing a sequence of actions that surpasses the length of the reference path annotated by human experts.

**Baselines** We consider the following baselines: **1. Episodic Transformer (E.T.)** (Pashevich et al., 2021) is an end-to-end multimodal transformer that encodes language inputs and a history of visual observations to predict actions, trained with imitation learning from human demonstrations. **2. Jarvis** (Zheng et al., 2022) trains an LLM on the TEACh dialogue to generate high-level subgoals that mimic those performed by the human demonstrator. Jarvis uses a semantic map and the Episodic Transformer for object search. **3. FILM** (Min et al., 2021, 2022) fine-tunes an LLM to produce parametrized plan templates. Similar to Jarvis, FILM uses a semantic map for carrying out subgoals and a semantic policy for object search. **4. DANLI** (Zhang et al., 2022) fine-tunes an LLM to predict high-level subgoals, and uses symbolic planning over an object state and spatial map to create an execution plan. DANLI uses an object search module and manually-defined error correction.

HELPER differs from the baselines in its use of memory-augmented context-dependent prompting of pretrained LLMs and pretrained visual-language models for planning, failure diagnosis and recovery, and object search. We provide a more in-depth comparison of HELPER to previous work in Table 1.

**Evaluation** We show quantitative results for HELPER and the baselines on the TEACh Trajectory from Dialogue (TfD) and Execution from Di-

alogue History (EDH) validation split in Table 2. **On the TfD validation unseen, HELPER achieves a 13.73% task success rate and 14.17% goal-condition success rate, a relative improvement of 1.7x and 2.1x, respectively, over DANLI, the prior SOTA in this setting. HELPER additionally sets a new SOTA in the EDH task, achieving a 17.40% task success rate and 25.86% goal-condition success rate on validation unseen.**

### 4.2 Ablations

We ablate components of HELPER in order to quantify what matters for performance in Table 2 *Ablations*. We perform all ablations on the TEACh TfD validation unseen split. We draw the following conclusions:

**1. Retrieval-augmented prompting helps** for planning, re-planning and failure recovery. Replacing the memory-augmented prompts with a fixed prompt (w/o Mem Aug; Table 2) led to a relative 18% reduction in success rate.

**2. VLM error correction helps** the agent recover from failures. Removal of the visually-grounded plan correction (w/o Correction; Table 2) led to a relative 6% reduction in success rate.

**3. The pre-condition check and the LLM search help.** Removal of the action pre-condition checks (w/o Pre Check; Table 2) led to a relative 16% reduction in success rate. Replacing the LOCATOR LLM-based search with a random search (w/o LOCATOR; Table 2) led to a relative 12% reduction in success rate.

**4. Larger LLMs perform better.** Using GPT-3.5 (w GPT-3.5; Table 2) exhibits a relative 31% reduction in success rate compared to using GPT-4. Our findings on GPT-4's superior planning abilities align with similar findings from recent studies of Wu et al. (2023d); Bubeck et al. (2023); Liu et al. (2023a); Wang et al. (2023a).

**5. Perception is a bottleneck.** Using GT depth

Table 2: **Trajectory from Dialogue (TfD) and Execution from Dialog History (EDH) evaluation on the TEACh validation set.** Trajectory length weighted metrics are included in ( parentheses ). SR = success rate. GC = goal condition success rate.

| | TfD | | | | EDH | | | |
|---|---|---|---|---|---|---|---|---|
| | Unseen | | Seen | | Unseen | | Seen | |
| | SR | GC | SR | GC | SR | GC | SR | GC |
| E.T. | 0.48 (0.12) | 0.35 (0.59) | 1.02 (0.17) | 1.42 (4.82) | 7.8 (0.9) | 9.1 (1.7) | 10.2 (1.7) | 15.7 (4.1) |
| JARVIS | 1.80 (0.30) | 3.10 (1.60) | 1.70 (0.20) | 5.40 (4.50) | 15.80 (2.60) | 16.60 (8.20) | 15.10 (3.30) | 22.60 (8.70) |
| FILM | 2.9 (1.0) | 6.1 (2.5) | 5.5 (2.6) | 5.8 (11.6) | 10.2 (1.0) | 18.3 (2.7) | 14.3 (2.1) | 26.4 (5.6) |
| DANLI | 7.98 (3.20) | 6.79 (6.57) | 4.97 (1.86) | 10.50 (10.27) | 16.98 (7.24) | 23.44 (19.95) | 17.76 (9.28) | 24.93 (22.20) |
| HELPER (ours) | **13.73** (1.61) | **14.17** (4.56) | **12.15** (1.79) | **18.62** (9.28) | **17.40** (2.91) | **25.86** (7.90) | **18.59** (4.00) | **32.09** (9.81) |

*Ablations*

| | | |
|---|---|---|
| w/o Mem Aug | 11.27 (1.39) | 11.09 (4.00) |
| w/o Pre Check | 11.6 (1.36) | 11.32 (4.15) |
| w/o Correction | 12.9 (1.53) | 12.45 (4.91) |
| w/o LOCATOR | 12.09 (1.29) | 10.89 (3.83) |
| w/ GPT-3.5 | 9.48 (1.21) | 10.05 (3.68) |
| w/ GT depth | 15.85 (2.85) | 14.49 (6.89) |
| w/ GT depth,seg | 22.55 (6.39) | 30.00 (14.56) |
| w/ GT percept | 30.23 (9.12) | 50.46 (20.24) |

*User Feedback*

| | | |
|---|---|---|
| w/ Feedback 1 | 16.34 (1.67) | 14.70 (4.69) |
| w/ Feedback 2 | 17.48 (1.97) | 14.93 (4.74) |
| w/ GT percept, Feedback 2 | 37.75 (10.96) | 56.77 (19.80) |

(w/ GT depth; Table 2) led to an improvement of 1.15x compared to using estimated depth from RGB. Notable is the 1.77x improvement in path-length weighted success when using GT depth. This change is due to lower accuracy for far depths in our depth estimation network lower, thereby causing the agent to spend more time mapping the environment and navigating noisy obstacle maps. Using lidar or better map estimation techniques could mitigate this issue.

Using ground truth segmentation masks and depth (w/ GT depth, seg; Table 2) improves task success and goal-conditioned task success by 1.64x and 2.11x, respectively. This shows the limitations of frame-based object detection and late fusion of detection responses over time. 3D scene representations that fuse features earlier across views may significantly improve 3D object detection. Using GT perception (w/ GT percept; Table 2), which includes depth, segmentation, action success, oracle failure feedback, and increased API failure limit (50), led to 2.20x and 3.56x improvement.

### 4.3 Eliciting Users' Feedback

We enable HELPER to elicit sparse user feedback by asking *"Is the task completed to your satisfaction? Did I miss anything?"* once it believes it has completed the task, as explained in Section 3.1.2. The user will then respond with steps missed by

HELPER, and HELPER will re-plan based on this feedback. As shown in in Table 2 *User Feedback*, asking for a user's feedback twice improves performance by 1.27x. Previous works do not explore this opportunity of eliciting human feedback partly due to the difficulty of interpreting it—being free-form language—which our work addresses.

### 4.4 Personalization

We evaluate HELPER's ability to retrieve user-specific routines, as well as on their ability to modify the retrieved routines, with one, two, or three modifications, as discussed in 3.1.1. For example, for three modifications we might instruct HELPER: "Make me a Dax sandwich with 1 slice of tomato, 2 lettuce leaves, and add a slice of bread".

**Dataset** The evaluation tests 10 user-specific plans for each modification category in five distinct tasks: MAKE A SANDWICH; PREPARE BREAKFAST; MAKE A SALAD; PLACE X ON Y; and CLEAN X. The evaluation contains 40 user requests. The complete list of user-specific plans and modification requests can be found in the Appendix, Section C.

**Evaluation** We report the success rate in Table 3. HELPER generates the correct personalized plan for all but three instances, out of 40 evaluation requests. This showcases the ability of HELPER to acquire, retrieve and adapt plans based on context and previous user interactions.

## 5 Limitations

Our model in its current form has the following limitations:

**1. Simplified failure detection.** The AI2-THOR simulator much simplifies action failure detection which our work and previous works exploit (Min et al., 2021; Inoue and Ohashi, 2022). In a more general setting, continuous progress monitoring

Table 3: **Evaluation of HELPER for user personalization.** Reported is success of generating the correct plan for 10 personalized plans for a request of the original plan without modifications, and one, two, or three modifications to the original plan. These experiments use the `text-davinci-003` model as the prompted LLM.

|                | Success |
|----------------|---------|
| Original Plan  | 100%    |
| One Change     | 100%    |
| Two Changes    | 80%     |
| Three Changes  | 90%     |

from pixels would be required for failure detection, which model VLMs can deliver and we will address in future work.

**2. 3D perception bottleneck.** HELPER relies on 2D object detectors and depth 3D lifting for 3D object localization. We observe a 2X boost in TEACh success rate from using ground truth segmentation in HELPER. In future work we plan to integrate early 2D features into persistent 3D scene feature representations for more accurate 3D object detection.

**4. Cost from LLM querying.** GPT-4 API is the most accurate LLM used in HELPER and incurs a significant cost. NLP research in model compression may help decreasing these costs, or finetuning smaller models with enough input-output pairs.

**3. Multimodal (vision and language) memory retrieval.** Currently, we use a text bottleneck in our environment state descriptions. Exciting future directions include exploration of visual state incorporation to the language model and partial adaptation of its parameters. A multi-modal approach to the memory and plan generation would help contextualize the planning more with the visual state.

Last, to follow human instructions outside of simulation environments our model would need to interface with robot closed-loop policies instead of abstract manipulation primitives, following previous work (Liang et al., 2022).

## 6   Conclusion

We presented HELPER, an instructable embodied agent that uses memory-augmented prompting of pre-trained LLMs to parse dialogue segments, instructions and corrections to programs over action primitives, that it executes in household environments from visual input. HELPER updates its memory with user-instructed action programs after successful execution, allowing personalized interactions by recalling and adapting them. It sets a new state-of-the-art in the TEACh benchmark. Future research directions include extending the model to include a visual modality by encoding visual context during memory retrieval or as direct input to the LLM. We believe our work contributes towards exciting new capabilities for instructable and conversable systems, for assisting users and personalizing human-robot communication.

## 7   Acknowledgements

This material is based upon work supported by National Science Foundation grants GRF DGE1745016 & DGE2140739 (GS), a DARPA Young Investigator Award, a NSF CAREER award, an AFOSR Young Investigator Award, and DARPA Machine Common Sense, and an ONR award AWD00002287 . Any opinions, findings and conclusions or recommendations expressed in this material are those of the authors and do not necessarily reflect the views of the United States Army, the National Science Foundation, or the United States Air Force.

This research project has benefitted from the Microsoft Accelerate Foundation Models Research (AFMR) grant program through which leading foundation models hosted by Microsoft Azure along with access to Azure credits were provided to conduct the research.

## Ethics Statement

The objective of this research is to construct autonomous agents. Despite the absence of human experimentation, practitioners could potentially implement this technology in human-inclusive environments. Therefore, applications of our research should appropriately address privacy considerations.

All the models developed in this study were trained using Ai2Thor (Kolve et al., 2017). Consequently, there might be an inherent bias towards North American homes. Additionally, we only consider English language inputs in this study.

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

## A  Analysis of User Personalization Failures

The three instances where the PLANNER made errors in the user personalization experiment (Section 4.4) involved logical mistakes or inappropriate alterations to parts of the original plan that were not requested for modification. For instance, in one case — involving a step to modify the plan from making one coffee to two coffees — the PLANNER includes placing two mugs in the coffee maker simultaneously, which is not a valid plan. In the other two instances, the PLANNER omits an object from the original plan that was not mentioned in the modification.

## B  User Feedback Details

In the user feedback evaluation, once the agent has indicated completion of the task from the original input dialogue, the agent will query feedback from the user. If the simulator indicates success of the task, the agent will end the episode. If the simulator indicates the task is not successful, feedback will be given to the agent for additional planning. This feedback is programatically generated from the TEACh simulator metadata, which gives us information about if the task is successful, and what object state changes are missing in order to complete the task (e.g., bread slice is not toasted, etc.). For each object state that is incorrect, we form a sentence of the following form: "You failed to complete the subtask: subtask. For the object object: description of desired object state." We combine all subtask sentences to create the feedback. HELPER follows the same pipeline (including examples, retrieval, planning, etc.) to process the feedback as with the input dialogue in the normal TfD evaluation. We show experiments with one and two user feedback requests in Section 4.3 of the main paper (a second request is queried if the first user feedback fails to produce task success).

## C  User Personalization Inputs

We provide a full list of the user personalization requests in Listing 1 for the user personalization experiments in Section 4.4.

## D  Prompts

We provide our full API (Listing 2), corrective API (Listing 3), PLANNER prompt (Listing 4), re-planning prompt (Listing 5), and LOCATOR prompt (Listing 6).

## E  Pre-conditions

An example of a pre-condition check for a macro-action is provided in Listing 7.

## F  Example LLM inputs & Outputs

We provide examples of dialogue input, retrieved examples, and LLM output for a TEACh sample in Listing 8, Listing 9, and Listing 10.

## G  Simulation environment

The TEACh dataset builds on the Ai2thor simulation environment (Kolve et al., 2017). At each time step the agent may choose from the following actions: Forward(), Backward(), Turn Left(), Turn Right(), Look Up(), Look Down(), Strafe Left(), Strafe Right(), Pickup(X), Place(X), Open(X), Close(X), ToggleOn(X), ToggleOff(X), Slice(X), and Pour(X), where X refers an object specified via a relative coordinate $(x, y)$ on the egocentric RGB frame. Navigation actions move the agent in discrete steps. We rotate in the yaw direction by 90 degrees, and rotate in the pitch direction by 30 degrees. The RGB and depth sensors are at a resolution of 480x480, a field of view of 90 degrees, and lie at a height of 0.9015 meters. The agent's coordinates are parameterized by a single $(x, y, z)$ coordinate triplet with $x$ and $z$ corresponding to movement in the horizontal plane and $y$ reserved for the vertical direction. The TEACh benchmark allows a maximum of 1000 steps and 30 API failures per episode.

## H  EXECUTOR details

### H.1  Semantic mapping and planning

**Obstacle map**  HELPER maintains a 2D overhead occupancy map of its environment $\in \mathbb{R}^{H \times W}$ that it updates at each time step from the input RGB-D stream. The map is used for exploration and navigation in the environment.

At every time step $t$, we unproject the input depth maps using intrinsic and extrinsic information of the camera to obtain a 3D occupancy map registered to the coordinate frame of the agent, similar to earlier navigation agents (Chaplot et al., 2020a). The 2D overhead maps of obstacles and free space are computed by projecting the 3D occupancy along the height direction at multiple height levels and summing. For each input RGB image,

we run a SOLQ object segmentor (Dong et al., 2021) (pretrained on COCO (Lin et al., 2014) then finetuned on TEACh rooms) to localize each of 116 semantic object categories. For failure detection, we use a simple matching approach from Min et al. (2021) to compare RGB pixel values before and after taking an action.

**Object location and state tracking** We maintain an object memory as a list of object detection 3D centroids and their predicted semantic labels $\{[(X, Y, Z)_i, \ell_i \in \{1...N\}], i = 1..K\}$, where $K$ is the number of objects detected thus far. The object centroids are expressed with respect to the coordinate system of the agent, and, similar to the semantic maps, updated over time using egomotion. We track previously detected objects by their 3D centroid $C \in \mathbb{R}^3$. We estimate the centroid by taking the 3D point corresponding to the median depth within the segmentation mask and bring it to a common coordinate frame. We do a simple form of non-maximum suppression on the object memory, by comparing the euclidean distance of centroids in the memory to new detected centroids of the same category, and keep the one with the highest score if they fall within a distance threshold.

For each object in the object memory, we maintain an object state dictionary with a pre-defined list of attributes. These attributes include: category label, centroid location, holding, detection score, can use, sliced, toasted, clean, cooked. For the binary attributes, these are initialized by sending the object crop, defined by the detector mask, to the VLM model, and checking its match to each of [f"The {object_category} is {attribute}", f"The {object_category} is not {attribute}"]. We found that initializing these attributes with the VLM gave only a marginal difference to initializing them to default values in the TEACh benchmark, so we do not use it for the TEACh evaluations. However, we anticipate a general method beyond dataset biases of TEACh would much benefit from such vision-based attribute classification.

**Exploration and path planning** HELPER explores the scene using a classical mapping method. We take the initial position of the agent to be the center coordinate in the map. We rotate the agent in-place and use the observations to instantiate an initial map. Second, the agent incrementally completes the maps by randomly sampling an unexplored, traversable location based on the 2D occu-

pancy map built so far, and then navigates to the sampled location, accumulating the new information into the maps at each time step. The number of observations collected at each point in the 2D occupancy map is thresholded to determine whether a given map location is explored or not. Unexplored positions are sampled until the environment has been fully explored, meaning that the number of unexplored points is fewer than a predefined threshold.

To navigate to a goal location, we compute the geodesic distance to the goal from all map locations using graph search (Inoue and Ohashi, 2022) given the top-down occupancy map and the goal location in the map. We then simulate action sequences and greedily take the action sequence which results in the largest reduction in geodesic distance.

### H.2   2D-to-3D unprojection

For the $i$-th view, a 2D pixel coordinate $(u, v)$ with depth $z$ is unprojected and transformed to its coordinate $(X, Y, Z)^T$ in the reference frame:

$$(X, Y, Z, 1) = \mathbf{G}_i^{-1} \left( z \frac{u - c_x}{f_x}, z \frac{v - c_y}{f_y}, z, 1 \right)^T \tag{1}$$

where $(f_x, f_y)$ and $(c_x, c_y)$ are the focal lengths and center of the pinhole camera model and $\mathbf{G}_i \in SE(3)$ is the camera pose for view $i$ relative to the reference view. This module unprojects each depth image $I_i \in \mathbb{R}^{H \times W \times 3}$ into a pointcloud in the reference frame $P_i \in \mathbb{R}^{M_i \times 3}$ with $M_i$ being the number of pixels with an associated depth value.

## I   Additional Experiments

### I.1   Alternate EDH Evaluation Split

Currently, the leaderboard for the TEACh EDH benchmark is not active. Thus, we are not able to evaluate on the true test set for TEACh. We used the original validation seen and unseen splits, which have been used in most previous works (Pashevich et al., 2021; Zheng et al., 2022; Min et al., 2022; Zhang et al., 2022). In Table 4 we report the alternative validation and test split as mentioned in the TEACh github README, and also reported by DANLI (Zhang et al., 2022).

Table 4: **Alternative TEACh Execution from Dialog History (EDH) evaluation split.** Trajectory length weighted metrics are included in ( parentheses ). SR = success rate. GC = goal condition success rate. Note that Test Seen and Unseen are not the true TEACh test sets, but an alternative split of the validation set used until the true test evaluation is released, as mentioned in the TEACh github README, and also reported by DANLI (Zhang et al., 2022).

| | Validation | | | | Test | | | |
| | Unseen | | Seen | | Unseen | | Seen | |
| | SR | GC | SR | GC | SR | GC | SR | GC |
| --- | --- | --- | --- | --- | --- | --- | --- | --- |
| E.T. | 8.35 (0.86) | 6.34 (3.69) | 8.28 (1.13) | 8.72 (3.82) | 7.38 (0.97) | 6.06 (3.17) | 8.82 (0.29) | 9.46 (3.03) |
| DANLI | **17.25** (7.16) | 23.88 (19.38) | 16.89 (9.12) | 25.10 (22.56) | 16.71 (7.33) | 23.00 (20.55) | **18.63** (9.41) | 24.77 (21.90) |
| HELPER | **17.25** (3.22) | **25.24** (8.12) | **19.21** (4.72) | **33.54** (10.95) | **17.55** (2.59) | **26.49** (7.67) | 17.97 (3.44) | **30.81** (8.93) |

Listing 1: Full list of user personalization requests for the user personalization evaluation.

```
original input to LLM:
[['Driver', 'What is my task?'], ['Commander', "Make me a sandwich. The name of this sandwich is
    called the Larry sandwich. The sandwich has two slices of toast, 3 slices of tomato, and 3
    slice of lettuce on a clean plate."]]
[['Driver', 'What is my task?'], ['Commander', 'Make me a salad. The name of this salad is called
    the David salad. The salad has two slices of tomato and three slices of lettuce on a clean
    plate.']]
[['Driver', 'What is my task?'], ['Commander', "Make me a salad. The name of this salad is called
    the Dax salad. The salad has two slices of cooked potato. You'll need to cook the potato on the
    stove. The salad also has a slice of lettuce and a slice of tomato. Put all components on a
    clean plate."]]
[['Driver', 'What is my task?'], ['Commander', 'Make me breakfast. The name of this breakfast is
    called the Mary breakfast. The breakfast has a mug of coffee, and two slices of toast on a
    clean plate.']]
[['Driver', 'What is my task?'], ['Commander', 'Make me breakfast. The name of this breakfast is
    called the Lion breakfast. The breakfast has a mug of coffee, and four slices of tomato on a
    clean plate.']]
[['Driver', 'What is my task?'], ['Commander', 'Rearrange some objects. The name of this
    rearrangement is called the Lax rearrangement. Place three pillows on the sofa.']]
[['Driver', 'What is my task?'], ['Commander', 'Rearrange some objects. The name of this
    rearrangement is called the Pax rearrangement. Place two pencils and two pens on the desk.']]
[['Driver', 'What is my task?'], ['Commander', 'Clean some objects. The name of this cleaning is
    called the Gax cleaning. Clean two plates and two cups.']]
[['Driver', 'What is my task?'], ['Commander', "Make me a sandwich. The name of this sandwich is
    called the Gabe sandwich. The sandwich has two slices of toast, 2 slices of tomato, and 1 slice
    of lettuce on a clean plate."]]
[['Driver', 'What is my task?'], ['Commander', 'Clean some objects. The name of this cleaning is
    called the Kax cleaning. Clean a mug and a pan.']]

No change:
"Make me the Larry sandwich"
"Make me the David salad"
"Make me the Dax salad"
"Make me the Mary breakfast"
"Make me the Lion breakfast"
"Complete the Lax rearrangement"
"Complete the Pax rearrangement"
"Perform the Gax cleaning"
"Make me the Gabe sandwich"
"Perform the Kax cleaning"

One change:
"Make me the Larry sandwich with four slices of lettuce"
"Make me the David salad with a slice of potato"
"Make me the Dax salad without lettuce"
"Make me the Mary breakfast with no coffee"
"Make me the Lion breakfast with three slice of tomato"
"Complete the Lax rearrangement with two pillows"
"Complete the Pax rearrangement but use one pencil instead of the the two pencils"
"Perform the Gax cleaning with three plates instead of two"
"Make me the Gabe sandwich with only 1 slice of tomato"
"Perform the Kax cleaning with only a mug"
```

"Make me the Larry sandwich with four slices of lettuce and two slices of tomato"
"Make me the David salad but add a slice of potato and add one slice of egg"
"Make me the Dax salad without lettuce and without potato"
"Make me the Mary breakfast with no coffee and add an egg"
"Make me the Lion breakfast with three slice of tomato and two mugs of coffee"
"Complete the Lax rearrangement with two pillows and add a remote"
"Complete the Pax rearrangement but use one pencil instead of the two pencils and add a book"
"Perform the Gax cleaning with three plates instead of the two plates and include a fork"
"Make me the Gabe sandwich with only 1 slice of tomato and two slices of lettuce"
"Perform the Kax cleaning without the pan and include a spoon"

"Make me the Larry sandwich with four slices of lettuce, two slices of tomato, and place all
     components directly on the countertop"
"Make me the David salad and add a slice of potato, add one slice of egg, and bring a fork with it"
"Make me the Dax salad without lettuce, without potato, and add an extra slice of tomato"
"Make me the Mary breakfast with no coffee, add an egg, and add a cup filled with water"
"Make me the Lion breakfast with three slice of tomato, two mugs of coffee, and add a fork"
"Complete the Lax rearrangement with two pillows, a remote, and place it on the arm chair instead"
"Complete the Pax rearrangement but use one pencil instead of the two pencils and include a book and
     a baseball bat"
"Perform the Gax cleaning with three plates instead of the two plates, include a fork, and do not
     clean any cups"
"Make me the Gabe sandwich with only 1 slice of tomato, two slices of lettuce, and add a slice of
     egg"
"Perform the Kax cleaning without the pan, include a spoon, and include a pot"

Listing 2: Full API for the parametrized macro-actions $G$ used in the prompts.

```python
class InteractionObject:
    """
    This class represents an expression that uniquely identifies an object in the house.
    """
    def __init__(self, object_class: str, landmark: str = None, attributes: list = []):
        '''
        object_class: object category of the interaction object (e.g., "Mug", "Apple")
        landmark: (optional if mentioned) landmark object category that the interaction object is in
            relation to (e.g., "CounterTop" for "apple is on the countertop")
        attributes: (optional) list of strings of desired attributes for the object. These are not
            necessarily attributes that currently exist, but ones that the object should eventually
            have. Attributes can only be from the following: "toasted", "clean", "cooked"
        '''
        self.object_class = object_class
        self.landmark = landmark
        self.attributes = attributes

    def pickup(self):
        """pickup the object.

        This function assumes the object is in view.

        Example:
        dialogue: <Commander> Go get the lettuce on the kitchen counter.
        Python script:
        target_lettuce = InteractionObject("Lettuce", landmark = "CounterTop")
        target_lettuce.go_to()
        target_lettuce.pickup()
        """
        pass

    def place(self, landmark_name):
        """put the interaction object on the landmark_name object.

        landmark_name must be a class InteractionObject instance

        This function assumes the robot has picked up an object and the landmark object is in view.

        Example:
        dialogue: <Commander> Put the lettuce on the kitchen counter.
        Python script:
        target_lettuce = InteractionObject("Lettuce", landmark = "CounterTop")
        target_lettuce.go_to()
        target_lettuce.pickup()
        target_countertop = InteractionObject("CounterTop")
        target_countertop.go_to()
        target_lettuce.place(target_countertop)
        """
        pass

    def slice(self):
        """slice the object into pieces.

        This function assumes the agent is holding a knife and the agent has navigated to the object
            using go_to().

        Example:
        dialogue: <Commander> Cut the apple on the kitchen counter.
        Python script:
        target_knife = InteractionObject("Knife") # first we need a knife to slice the apple with
        target_knife.go_to()
        target_knife.pickup()
        target_apple = InteractionObject("Apple", landmark = "CounterTop")
        target_apple.go_to()
        target_apple.slice()
        """
        pass
```

```python
    def toggle_on(self):
        """toggles on the interaction object.

        This function assumes the interaction object is already off and the agent has navigated to
            the object.
        Only some landmark objects can be toggled on. Lamps, stoves, and microwaves are some
            examples of objects that can be toggled on.

        Example:
        dialogue: <Commander> Turn on the lamp.
        Python script:
        target_floorlamp = InteractionObject("FloorLamp")
        target_floorlamp.go_to()
        target_floorlamp.toggle_on()
        """
        pass

    def toggle_off(self):
        """toggles off the interaction object.

        This function assumes the interaction object is already on and the agent has navigated to
            the object.
        Only some objects can be toggled off. Lamps, stoves, and microwaves are some examples of
            objects that can be toggled off.

        Example:
        dialogue: <Commander> Turn off the lamp.
        Python script:
        target_floorlamp = InteractionObject("FloorLamp")
        target_floorlamp.go_to()
        target_floorlamp.toggle_off()
        """
        pass

    def go_to(self):
        """Navigate to the object

        """
        pass

    def open(self):
        """open the interaction object.

        This function assumes the landmark object is already closed and the agent has already
            navigated to the object.
        Only some objects can be opened. Fridges, cabinets, and drawers are some example of objects
            that can be closed.

        Example:
        dialogue: <Commander> Get the lettuce in the fridge.
        Python script:
        target_fridge = InteractionObject("Fridge")
        target_lettuce = InteractionObject("Lettuce", landmark = "Fridge")
        target_fridge.go_to()
        target_fridge.open()
        target_lettuce.pickup()
        """
        pass

    def close(self):
        """close the interaction object.

        This function assumes the object is already open and the agent has already navigated to the
            object.
        Only some objects can be closed. Fridges, cabinets, and drawers are some example of objects
            that can be closed.
        """
        pass

    def clean(self):
```

```python
        """wash the interaction object to clean it in the sink.

        This function assumes the object is already picked up.

        Example:
        dialogue: <Commander> Clean the bowl
        Python script:
        target_bowl = InteractionObject("Bowl", attributes = ["clean"])
        target_bowl.clean()
        """
        pass

    def put_down(self):
        """puts the interaction object currently in the agent's hand on the nearest available
            receptacle

        This function assumes the object is already picked up.
        This function is most often used when the holding object is no longer needed, and the agent
            needs to pick up another object
        """
        pass

    def pour(self, landmark_name):
        """pours the contents of the interaction object into the landmark object specified by the
            landmark_name argument

        landmark_name must be a class InteractionObject instance

        This function assumes the object is already picked up and the object is filled with liquid.
        """
        pass

    def fill_up(self):
        """fill up the interaction object with water

        This function assumes the object is already picked up. Note that only container objects can
            be filled with liquid.
        """
        pass

    def pickup_and_place(self, landmark_name):
        """go_to() and pickup() this interaction object, then go_to() and place() the interaction
            object on the landmark_name object.

        landmark_name must be a class InteractionObject instance
        """
        pass

    def empty(self):
        """Empty the object of any other objects on/in it to clear it out.

        Useful when the object is too full to place an object inside it.

        Example:
        dialogue: <Commander> Clear out the sink.
        Python script:
        target_sink = InteractionObject("Sink")
        target_sink.empty()
        """
        pass

    def cook(self):
        """Cook the object

        Example:
        dialogue: <Commander> Cook the potato.
        Python script:
        target_potato = InteractionObject("Potato", attributes = ["cooked"])
        target_potato.cook()
        """
```

```python
        pass

    def toast(self):
        """Toast a bread slice in a toaster

        Toasting is only supported with slices of bread

        Example:
        dialogue: <Commander> Get me a toasted bread slice.
        Python script:
        target_breadslice = InteractionObject("BreadSliced", attributes = ["toasted"])
        target_breadslice.toast()
        """
        pass
```

Listing 3: Full Corrective API for the parametrized corrective macro-actions $G_{corrective}$ used in the prompts.

```python
class AgentCorrective:
    '''
    This class represents agent corrective actions that can be taken to fix a subgoal error
    Example usage:
    agent = AgentCorrective()
    agent.move_back()
    '''

    def move_back(self):
        """Step backwards away from the object

        Useful when the object is too close for the agent to interact with it
        """
        pass

    def move_closer(self):
        """Step forward to towards the object to get closer to it

        Useful when the object is too far for the agent to interact with it
        """
        pass

    def move_alternate_viewpoint(self):
        """Move to an alternate viewpoint to look at the object

        Useful when the object is occluded or an interaction is failing due to collision or
            occlusion.
        """
        pass
```

Listing 4: Full Prompt for the PLANNER. {} indicates areas that are replaced in the prompt.

You are an adept at translating human dialogues into sequences of actions for household robots.
  Given a dialogue between a <Driver> and a <Commander>, you convert the conversation into a
  Python program to be executed by a robot.

{API}

Write a script using Python and the InteractionObject class and functions defined above that could
  be executed by a household robot.

{RETRIEVED_EXAMPLES}

Adhere to these stringent guidelines:
1. Use only the classes and functions defined previously. Do not create functions that are not
  provided above.
2. Make sure that you output a consistent plan. For example, opening of the same object should not
  occur in successive steps.
3. Make sure the output is consistent with the proper affordances of objects. For example, a couch
  cannot be opened, so your output should never include the open() function for this object, but
  a fridge can be opened.
4. The input is dialogue between <Driver> and <Commander>. Interpret the dialogue into robot
  actions. Do not output any dialogue.
5. Object categories should only be chosen from the following classes: ShowerDoor, Cabinet,
  CounterTop, Sink, Towel, HandTowel, TowelHolder, SoapBar, ToiletPaper, ToiletPaperHanger,
  HandTowelHolder, SoapBottle, GarbageCan, Candle, ScrubBrush, Plunger, SinkBasin, Cloth,
  SprayBottle, Toilet, Faucet, ShowerHead, Box, Bed, Book, DeskLamp, BasketBall, Pen, Pillow,
  Pencil, CellPhone, KeyChain, Painting, CreditCard, AlarmClock, CD, Laptop, Drawer, SideTable,
  Chair, Blinds, Desk, Curtains, Dresser, Watch, Television, WateringCan, Newspaper, FloorLamp,
  RemoteControl, HousePlant, Statue, Ottoman, ArmChair, Sofa, DogBed, BaseballBat, TennisRacket,
  VacuumCleaner, Mug, ShelvingUnit, Shelf, StoveBurner, Apple, Lettuce, Bottle, Egg, Microwave,
  CoffeeMachine, Fork, Fridge, WineBottle, Spatula, Bread, Tomato, Pan, Cup, Pot, SaltShaker,
  Potato, PepperShaker, ButterKnife, StoveKnob, Toaster, DishSponge, Spoon, Plate, Knife,
  DiningTable, Bowl, LaundryHamper, Vase, Stool, CoffeeTable, Poster, Bathtub, TissueBox,
  Footstool, BathtubBasin, ShowerCurtain, TVStand, Boots, RoomDecor, PaperTowelRoll, Ladle,
  Kettle, Safe, GarbageBag, TeddyBear, TableTopDecor, Dumbbell, Desktop, AluminumFoil, Window,
  LightSwitch, AppleSliced, BreadSliced, LettuceSliced, PotatoSliced, TomatoSliced
6. You can only pick up one object at a time. If the agent is holding an object, the agent should
  place or put down the object before attempting to pick up a second object.
7. Each object instance should instantiate a different InteractionObject class even if two object
  instances are the same object category.
Follow the output format provided earlier. Think step by step to carry out the instruction.

Write a Python script that could be executed by a household robot for the following:
dialogue: {command}
Python script:

Listing 5: Full Prompt for the RECTIFIER. {} indicates areas that are replaced in the prompt.

You are an excellent interpreter of human instructions for household tasks. Given a failed action
    subgoal by a household robot, dialogue instructions between robot <Driver> and user
    <Commander>, and information about the environment and failure, you provide a sequence of
    robotic subgoal actions to overcome the failure.

{API}

{API_CORRECTIVE}

Information about the failure and environment are given as follows:
Failed subgoal: The robotic subgoal for which the failure occured.
Execution error: feedback as to why the failed subgoal occurred.
Input dialogue: full dialogue instructions between robot <Driver> and user <Commander> for the
    complete task. This may or may not be useful.

I will give you examples of the input and output you will generate.
{retrieved_plans}

Fix the subgoal exectuion error using only the InteractionObject class and functions defined above
    that could be executed by a household robot. Follow these rules very strictly:
1. Important! Use only the classes and functions defined previously. Do not create functions or
    additional code that are not provided in the above API. Do not include if-else statements.
2. Important! Make sure that you output a consistent plan. For example, opening of the same object
    should not occur in successive steps.
3. Important! Make sure the output is consistent with the proper affordances of objects. For
    example, a couch cannot be opened, so your output should never include the open() function for
    this object, but a fridge can be opened.
4. Important! The dialogue is between <Driver> and <Commander>. The dialogue may or may not be
    helpful. Do not output any dialogue.
5. Important! Object classes should only be chosen from the following classes: ShowerDoor, Cabinet,
    CounterTop, Sink, Towel, HandTowel, TowelHolder, SoapBar, ToiletPaper, ToiletPaperHanger,
    HandTowelHolder, SoapBottle, GarbageCan, Candle, ScrubBrush, Plunger, SinkBasin, Cloth,
    SprayBottle, Toilet, Faucet, ShowerHead, Box, Bed, Book, DeskLamp, BasketBall, Pen, Pillow,
    Pencil, CellPhone, KeyChain, Painting, CreditCard, AlarmClock, CD, Laptop, Drawer, SideTable,
    Chair, Blinds, Desk, Curtains, Dresser, Watch, Television, WateringCan, Newspaper, FloorLamp,
    RemoteControl, HousePlant, Statue, Ottoman, ArmChair, Sofa, DogBed, BaseballBat, TennisRacket,
    VacuumCleaner, Mug, ShelvingUnit, Shelf, StoveBurner, Apple, Lettuce, Bottle, Egg, Microwave,
    CoffeeMachine, Fork, Fridge, WineBottle, Spatula, Bread, Tomato, Pan, Cup, Pot, SaltShaker,
    Potato, PepperShaker, ButterKnife, StoveKnob, Toaster, DishSponge, Spoon, Plate, Knife,
    DiningTable, Bowl, LaundryHamper, Vase, Stool, CoffeeTable, Poster, Bathtub, TissueBox,
    Footstool, BathtubBasin, ShowerCurtain, TVStand, Boots, RoomDecor, PaperTowelRoll, Ladle,
    Kettle, Safe, GarbageBag, TeddyBear, TableTopDecor, Dumbbell, Desktop, AluminumFoil, Window,
    LightSwitch, AppleSliced, BreadSliced, LettuceSliced, PotatoSliced, TomatoSliced
6. Important! You can only pick up one object at a time. If the agent is holding an object, the
    agent should place or put down the object before attempting to pick up a second object.
7. Important! Each object instance should instantiate a different InteractionObject class even if
    two object instances are the same object category.
8. Important! Your plan should ONLY fix the failed subgoal. Do not include plans for other parts of
    the dialogue or future plan that are irrelevant to the execution error and failed subgoal.
9. Important! output "do_nothing()" if the agent should not take any corrective actions.
Adhere to the output format I defined above. Think step by step to carry out the instruction.

Make use of the following information to help you fix the failed subgoal:
Failed subgoal: ...
Execution error: ...
Input dialogue: ...

You should respond in the following format:
Explain: Are there any steps missing to complete the subgoal? Why did the failed subgoal occur? What
    does the execution error imply for how to fix your future plan?
Plan (Python script): A Python script to only fix the execution error.

Explain:

Listing 6: Full Prompt for the LOCATOR. {} indicates areas that are replaced in the prompt.

```
You are a household robot trying to locate objects within a house.
You will be given a target object category, your task is to output the top 3 most likely object
    categories that the target object category is likely to be found near: {OBJECT_CLASSES}
For your answer, take into account commonsense co-occurances of objects within a house and (if
    relevant) any hints given by the instruction dialogue between the robot <Driver> and user
    <Commander>.

For example, if given the target object category is "Knife" and the following dialogue: "<Commander>
    hi, make a slice of tomato. <Driver> where is the tomato? <Driver> where is the knife?
    <Commander> in the sink.", you might output the following top 3 most likely object categories:
    "answer: Sink, CounterTop, Cabinet". Important: Your output should ONLY a list (3 words
    seperated by commas) of three object categories from the list above.

What are the top 3 most likely object categories for where to find the target category
    {INPUT_TARGET_OBJECT} near? Here is the dialogue that may or may not contain relevant
    information for your answer: {INPUT_DIALOGUE}

answer:
```

Listing 7: Example of a precondition check for the slice macro-action

```python
if subgoal=="Slice":
    # contraint check for "slice" macro-action
    if object_tracker.get_label_of_holding()!="Knife":
        # first need to be holding a knife to slice
        program_to_add = [
            'target_knife = InteractionObject("Knife")',
            'target_knife.go_to()',
            'target_knife.pickup()',
        ]
```

**Listing 8: Example of dialogue input, retrieved examples, and LLM output for a TEACh sample**

```
<Driver> how can I help you today?<Commander> can you please make me a salad on a clean plate with
    tomato and cooked potato?<Driver> does the salad require chopped lettuce?<Commander> nope!<Driver>
    is that all?<Commander> can you place them on a plate?<Driver> are they not already on a plate?
```

Retrieved Examples:
Example 1:
```
dialogue: <Driver> What should I do today? <Commander> hi, make a slice of tomato. <Driver> where is
    the tomato? <Driver> where is the knife? <Commander> in the sink. <Driver> Tomato sliced. What
    next? <Commander> slice the potato. <Driver> Where is the potato? <Commander> in the microwave.
    <Commander> place all salad components on a plate. <Driver> How many slices of potato?
    <Commander> all salad components need to be place on a plate. <Driver> Where is the plate?
    <Commander> plate. <Commander> try the tomato piece. <Driver> done. <Commander> we are finished.
```
```python
Python script:
target_tomato = InteractionObject("Tomato")
target_knife = InteractionObject("Knife") # need knife to slice tomato
target_knife.go_to() # first get knife
target_knife.pickup()
target_tomato.go_to()
target_tomato.slice()
target_tomato_slice1 = InteractionObject("TomatoSliced") # instantiate a tomato slice object after
    slicing tomato
target_potato = InteractionObject("Potato", landmark = "Microwave")
target_potato.go_to()
target_potato.slice()
target_potato_slice1 = InteractionObject("PotatoSliced") # instantiate a potato slice object after
    slicing potato
target_knife.put_down() # put down the knife to free up the agent's hand
target_plate = InteractionObject("Plate")
target_tomato_slice1.pickup_and_place(target_plate) # pick up the tomato slice and place it on the
    plate
target_potato_slice1.pickup_and_place(target_plate) # pick up the potato slice and place it on the
    plate
```

Example 2:
```
dialogue: <Driver> how can I help? <Commander> please serve 4 slices of tomato on a plate. <Driver>
    sure. <Driver> where can i find the tomato? <Driver> an knife. <Commander> Tomato on countertop
    and knife in fridge. <Commander> there should be a plate on the right cupboard by the sink.
    <Driver> completed. <Commander> great work thanks.
```
```python
Python script:
target_tomato = InteractionObject("Tomato")
target_knife = InteractionObject("Knife") # need knife to slice tomato
target_knife.go_to() # first get knife
target_knife.pickup()
target_tomato.go_to()
target_tomato.slice()
target_plate = InteractionObject("Plate", landmark = "Sink")
# place 4 slices of the cut tomato on the plate
target_tomato_slice1 = InteractionObject("TomatoSliced") # instantiate first slice of tomato
target_tomato_slice2 = InteractionObject("TomatoSliced") # instantiate second slice of tomato
target_tomato_slice3 = InteractionObject("TomatoSliced") # instantiate third slice of tomato
target_tomato_slice4 = InteractionObject("TomatoSliced") # instantiate fourth slice of tomato
target_tomato_slice1.pickup_and_place(target_plate) # pick up the first tomato slice and place it on
    the plate
target_tomato_slice2.pickup_and_place(target_plate) # pick up the second tomato slice and place it
    on the plate
target_tomato_slice3.pickup_and_place(target_plate) # pick up the third tomato slice and place it on
    the plate
target_tomato_slice4.pickup_and_place(target_plate) # pick up the fourth tomato slice and place it
    on the plate
```

Example 3:
```
dialogue: <Driver> hi, what is task. <Commander> make a plate of toast. <Commander> bread in the sink.
    <Driver> knife. <Commander> on the chair beside the table. <Commander> clean the plate. <Driver>
    there is a clean one, do i need to clean the other one. <Commander> you can use the clean one.
    <Driver> done. <Commander> good job. <Driver> :).
```
```python
Python script:
target_bread = InteractionObject("Bread", landmark = "Sink")
target_knife = InteractionObject("Knife", landmark = "Chair")
target_toaster = InteractionObject("Toaster") # need knife to slice bread
target_knife.go_to() # first get knife
```

```
target_knife.pickup()
target_bread.go_to()
target_bread.slice() # slice bread using knife
target_knife.put_down() # put down the knife to free up the agent's hand
target_bread_slice1 = InteractionObject("BreadSliced") # instantiate a bread slice object after
    slicing bread
target_bread_slice1.pickup_and_place(target_toaster) # pick up the bread slice and place it in the
    toaster
target_toaster.toggle_on() # turn on toaster to toast bread
target_toaster.toggle_off() # turn off toaster to complete toasting process
# place toast on plate
target_plate = InteractionObject("Plate")
target_plate.clean()
target_plate.put_down()
target_bread_slice1.pickup_and_place(target_plate) # pick up the toasted bread slice and place it on
    the plate

LLM output:
target_tomato = InteractionObject("Tomato")
target_potato = InteractionObject("Potato")
target_plate = InteractionObject("Plate")
target_stove = InteractionObject("StoveBurner")

# Clean the plate
target_plate.go_to()
target_plate.clean()

# Cook the potato
target_potato.go_to()
target_potato.pickup()
target_stove.go_to()
target_potato.place(target_stove)
target_stove.toggle_on()
target_stove.toggle_off()

# Place the cooked potato on the plate
target_potato = InteractionObject("Potato")
target_potato.pickup()
target_plate.go_to()
target_potato.place(target_plate)

# Place the tomato on the plate
target_tomato.go_to()
target_tomato.pickup()
target_plate.go_to()
target_tomato.place(target_plate)
```

Listing 9: Example of dialogue input, retrieved examples, and LLM output for a TEACh sample

Dialogue input:
```
<Driver> what do i do today<Commander> Hi. Please place two tissue boxes on a table. One is on the
    gray chair in the corner.<Commander> The other is under the tv<Commander> tv*<Driver> where is
    the other one
```

Retrieved Examples:
Example 1:
```
dialogue: <Driver> hi, what is task. <Commander> make a plate of toast. <Commander> bread in the sink.
    <Driver> knife. <Commander> on the chair beside the table. <Commander> clean the plate. <Driver>
    there is a clean one, do i need to clean the other one. <Commander> you can use the clean one.
    <Driver> done. <Commander> good job. <Driver> :).
Python script:
target_bread = InteractionObject("Bread", landmark = "Sink")
target_knife = InteractionObject("Knife", landmark = "Chair")
target_toaster = InteractionObject("Toaster") # need knife to slice bread
target_knife.go_to() # first get knife
target_knife.pickup()
target_bread.go_to()
target_bread.slice() # slice bread using knife
target_knife.put_down() # put down the knife to free up the agent's hand
target_bread_slice1 = InteractionObject("BreadSliced") # instantiate a bread slice object after
    slicing bread
target_bread_slice1.pickup_and_place(target_toaster) # pick up the bread slice and place it in the
    toaster
target_toaster.toggle_on() # turn on toaster to toast bread
target_toaster.toggle_off() # turn off toaster to complete toasting process
# place toast on plate
target_plate = InteractionObject("Plate")
target_plate.clean()
target_plate.put_down()
target_bread_slice1.pickup_and_place(target_plate) # pick up the toasted bread slice and place it on
    the plate
```

Example 2:
```
dialogue: <Driver> how can I help? <Commander> please serve 4 slices of tomato on a plate. <Driver>
    sure. <Driver> where can i find the tomato? <Driver> an knife. <Commander> Tomato on countertop
    and knife in fridge. <Commander> there should be a plate on the right cupboard by the sink.
    <Driver> completed. <Commander> great work thanks.
Python script:
target_tomato = InteractionObject("Tomato", landmark = "CounterTop")
target_knife = InteractionObject("Knife", landmark = "Fridge") # need knife to slice tomato
target_knife.go_to() # first get knife
target_knife.pickup()
target_tomato.go_to()
target_tomato.slice()
target_plate = InteractionObject("Plate", landmark = "Sink")
# place 4 slices of the cut tomato on the plate
target_tomato_slice1 = InteractionObject("TomatoSliced") # instantiate first slice of tomato
target_tomato_slice2 = InteractionObject("TomatoSliced") # instantiate second slice of tomato
target_tomato_slice3 = InteractionObject("TomatoSliced") # instantiate third slice of tomato
target_tomato_slice4 = InteractionObject("TomatoSliced") # instantiate fourth slice of tomato
target_tomato_slice1.pickup_and_place(target_plate) # pick up the first tomato slice and place it on
    the plate
target_tomato_slice2.pickup_and_place(target_plate) # pick up the second tomato slice and place it
    on the plate
target_tomato_slice3.pickup_and_place(target_plate) # pick up the third tomato slice and place it on
    the plate
target_tomato_slice4.pickup_and_place(target_plate) # pick up the fourth tomato slice and place it
    on the plate
```

Example 3:
```
dialogue: <Driver> How can I help? <Commander> put all the scrub brushes on the counter top. <Driver>
    where can I find them? <Commander> one is on top of the toilet. <Driver> there is only a cloth.
    <Driver> I put it on the counter top. <Driver> are there more? <Commander> can you try with the
    brush to the side of the toilet please? <Driver> okay.
Python script:
target_scrub_brush1 = InteractionObject("ScrubBrush", landmark = "Toilet")
target_counter_top = InteractionObject("CounterTop")
target_scrub_brush1.pickup_and_place(target_counter_top) # pick up the first ScrubBrush and place it
    on the CounterTop
```

```
target_scrub_brush2 = InteractionObject("ScrubBrush", landmark = "Toilet")
target_scrub_brush2.pickup_and_place(target_counter_top) # pick up the second ScrubBrush and place
    it on the CounterTop
```

LLM output:
```
target_tissue_box1 = InteractionObject("TissueBox", landmark = "Chair")
target_table = InteractionObject("Table")
target_tissue_box1.pickup_and_place(target_table) # pick up the first TissueBox and place it on the
    Table
target_tissue_box2 = InteractionObject("TissueBox", landmark = "TVStand")
target_tissue_box2.pickup_and_place(target_table) # pick up the second TissueBox and place it on the
    Table
```

Listing 10: Example of dialogue input, retrieved examples, and LLM output for a TEACh sample

Dialogue input:
```
<Driver> hi how can i help<Commander> Can you prepare me a coffee in a clean mug<Commander> There
    should be a mug on the white table<Commander> it might need cleaning first<Commander> that's
    great<Driver> are they not already on a plate?
```

Retrieved Examples:
Example 1:
```
dialogue: <Commander> Prepare coffee in a clean mug.
Python script:
target_mug = InteractionObject("Mug")
target_mug.go_to()
target_mug.pickup()
# clean mug in the sink
target_mug.clean()
target_coffee_machine = InteractionObject("CoffeeMachine")
target_coffee_machine.go_to()
target_mug.place(target_coffee_machine)
target_coffee_machine.toggle_on() # turn on coffee machine to make coffee
target_coffee_machine.toggle_off() # turn off coffee machine after done
```

Example 2:
```
dialogue: <Driver> what can i for you today? <Commander> could you boil a potato? <Driver> sure thing!
    <Driver> are there any pots? <Commander> pots for boiling potato? <Driver> yes. <Commander> just
    the one that you were holding earlier and a couple others of the same size. <Commander> does it
    not fit in the sink? <Driver> no. <Commander> see that mug of water next to the pot? <Driver> yes.
    <Commander> pour the mugs water into the pot. <Driver> thanks. <Commander> you're welcome!
    <Commander> excellent thank you!
Python script:
target_pot = InteractionObject("Pot")
target_mug = InteractionObject("Mug", landmark = "Pot")
target_mug.go_to()
target_mug.pickup()
target_mug.pour(target_pot)
target_mug.put_down() # put down mug on any location to free up agent's hand
target_potato = InteractionObject("Potato")
target_potato.pickup_and_place(target_pot) # pick up the potato and place it in the pot
target_stove_knob = InteractionObject("StoveKnob") # turn on the stove to boil potato by turning on
    the stove
target_stove_knob.go_to()
target_stove_knob.toggle_on() # boil water
target_stove_knob.toggle_off() # complete boiling process by turning off
```

Example 3:
```
dialogue: <Driver> hi, what is task. <Commander> make a plate of toast. <Commander> bread in the sink.
    <Driver> knife. <Commander> on the chair beside the table. <Commander> clean the plate. <Driver>
    there is a clean one, do i need to clean the other one. <Commander> you can use the clean one.
    <Driver> done. <Commander> good job. <Driver> :).
Python script:
target_bread = InteractionObject("Bread", landmark = "Sink")
target_knife = InteractionObject("Knife", landmark = "Chair")
target_toaster = InteractionObject("Toaster", "Toaster") # need knife to slice bread
target_knife.go_to() # first get knife
target_knife.pickup()
target_bread.go_to()
target_bread.slice() # slice bread using knife
target_knife.put_down() # put down the knife to free up the agent's hand
target_bread_slice1 = InteractionObject("BreadSliced") # instantiate a bread slice object after
    slicing bread
target_bread_slice1.pickup_and_place(target_toaster) # pick up the bread slice and place it in the
    toaster
target_toaster.toggle_on() # turn on toaster to toast bread
target_toaster.toggle_off() # turn off toaster to complete toasting process
# place toast on plate
target_plate = InteractionObject("Plate")
target_plate.clean()
target_plate.put_down()
target_bread_slice1.pickup_and_place(target_plate) # pick up the toasted bread slice and place it on
    the plate
```

```
LLM output:
target_table = InteractionObject("Table")
target_mug = InteractionObject("Mug", landmark = "Table")
target_mug.go_to()
target_mug.pickup()
# clean mug in the sink
target_sink = InteractionObject("Sink")
target_mug.place(target_sink)
target_mug.clean()
target_mug.pickup()
target_coffee_machine = InteractionObject("CoffeeMachine")
target_coffee_machine.go_to()
target_mug.place(target_coffee_machine)
target_coffee_machine.toggle_on() # turn on coffee machine to make coffee
target_coffee_machine.toggle_off() # turn off coffee machine after done
```