# OpenReview forum: "Open-Ended Instructable Embodied Agents with Memory-Augmented Large Language Models"
_EMNLP/2023/Conference — EMNLP 2023 Findings_

### Official Review · Reviewer_31Xh · 2023-07-26

**Soundness:** 4

**Excitement:**

4: Strong: This paper deepens the understanding of some phenomenon or lowers the barriers to an existing research direction.

**Paper Topic And Main Contributions:**

This paper presents HELPER, an embodied agent that uses LLMs to plan, correct errors, and personalize behaviors for instruction-following tasks. HELPER is set in the TEACh environment, which consists of dialogue-based interactive tasks. HELPER takes language descriptions of the current and past observations, and outputs API-level symbolic programs. To incorporate feedback, the agent retrieves the closest language-program pair from a prior dataset. Experiments on the TEACh benchmark indicate that HELPER substantially outperforms prior SoTA approaches by 1.4x.


**Reasons To Accept:**

+ HELPER is highly applicable to robotics and embodied-AI applications. Error correction and user personalization are common issues in robotics. Most attempts either hardcode solutions or rely on fixed databases with extensive human engineering. HELPER uses LLM prompting as a unified approach to tackle these issues.
+ HELPER achieves compelling quantitative results on the TEACh benchmark. HELPER improves over the previous best method by 1.4x on task success and 2.54x on goal-condition success (partial goals).
+ The experiments include a good set of baselines and ablations. Several models, i.e ET, JARVIS, FILM, and DANLI, are investigated. Table 2 includes extensive ablation experiments that investigate HELPER’s individual components.


**Reasons To Reject:**

- The overall organization of the paper could be greatly improved. There are at least 3 different problems tackled by HELPER: (1) dialogue grounding, (2) failure correction, and (3) user personalization. Perhaps before diving into methods, a unifying problem statement would be helpful, or at least a unifying example that motivates these problems, and why the paper chose to specifically tackle them.
- The frequent use of “visuomotor” is a bit of an overclaim. “Visuomotor” commonly refers to connecting perception and low-level motor skills (at least in robotics). But HELPER only outputs high-level API actions like “pickUp(apple)”. This comment may seem like an issue with word choice, but it hints at a deeper limitation in the use of symbolic action spaces. Certain actions like “pulling noodles” or “folding clothes” might be difficult to abstract and parameterize with symbolic actions.
- The evaluations are limited to TEACh. It seems like HELPER is generally applicable to other domains like navigation. Perhaps running some evaluations on CVDN (Thomason et al., 2019) and other vision-dialogue-tasks could show the generalizability of HELPER?

References:

Thomason et al 2019 – https://arxiv.org/abs/1907.04957

**Reproducibility:**

4: Could mostly reproduce the results, but there may be some variation because of sample variance or minor variations in their interpretation of the protocol or method.

**Reviewer Confidence:**

4: Quite sure. I tried to check the important points carefully. It's unlikely, though conceivable, that I missed something that should affect my ratings.

---

> ### Author Rebuttal · Authors · 2023-08-27
>
> Q: **The evaluations are limited to TEACh. What about evaluation on CVDN (Thomason et al., 2019).**
>
> A: Thank you for your suggestion. You are correct that our model can be easily generalized to other tasks, and we agree this would be useful to show. Per your  (and hr3J's) suggestion, we have evaluated  HELPER to the EDH task of TEACH.** For EDH, we provide the LLM with the subgoals completed thus far by the Driver in addition to the input dialogue and instruct the LLM to only output the remaining program to complete the task. We use the agent history provided by the EDH episode to initialize our semantic map and object memory.  **HELPER achieves 18.3% success and 31.5% goal condition success on validation seen**. Previous SOTA in this task is Danli (Zhang et. Al. 2022)  with  16.89% success and 25.10% goal condition success.
> HELPER not only outperforms the baselines, but it does so without any in domain training or finetuning on the TEACh dataset. Our focus was initially on the TfD task because of its increased complexity and similar task structure to a robot deployed in a real household. We will include these results in the camera ready.
>
> Thank you for bringing the CVDN task to our attention. We were not aware of this, and we think it would be a useful addition to the paper. The task involves a different simulation environment and thus we dis not have time to include these results in this rebuttal. We are working on completing this evaluation.
>
>
>
> Q: **The overall organization of the paper could be greatly improved. A unifying problem statement would be helpful.**
>
> A: Thank you for your valuable feedback on the paper's organization. To improve the writing clarity we will do the following:
>
> 1. We will include a unifying problem statement with an example in the introduction to improve the clarity of the problems and proposed solutions:
>
> > In a home setting, consider a robot that receives task instructions from a human user. These instructions, as depicted in Figure 1 (Left), may be in free form natural language, with omissions, as well as references to previous objects and events, as shown in Figure 1 (Right). To understand the user's intent and generate a correct action sequence, the robot can refer to past successful interactions that are contextually relevant to the current instruction. As it carries out the task, the robot may need to make real-time adjustments, such as when encountering challenges involving a cluttered placement area (Figure 1, Middle). To navigate these challenges, the robot should use visual feedback to understand why a particular sub-task failed and adjust its plan accordingly.
>
> 2. To  address your concerns about organization, we have added an algorithm box, a visualization to demonstrate the interaction of the modules, and the following text:
>
> > The HELPER system is an embodied agent designed to convert natural language instructions and RGB sensor data into a sequence of robot actions. The system has a modular organization outlined in Algorithm S1 and visualized in Figure S4. The PLANNER module uses nearest neighbor retrieval to obtain the most relevant task plans from a memory bank by matching language encoding of accompanying dialogue. The retrieved language-plan pairs are used as in-context examples inside the prompt for an LLM to transform user instructions or dialogue into an executable program. This generated program is then sent to the EXECUTOR module, which translates each program step into specific navigation and manipulation actions. To support this, the system uses RGB sensor data to produce segmentation and depth images at each time step. These images are used for object identification and for building a semantic map, which is subsequently used for object tracking and navigation. Before executing each subgoal, the INSPECTOR module verifies if the necessary preconditions for an action, such as the robot already holding an object, are met. If not, the plan is adjusted according to the current environmental and agent state. Should a subgoal involve an undetected object, the EXECUTOR calls on the LOCATOR module. This module utilizes previous user instructions and LLMs' commonsense knowledge to efficiently search for the required object. If any action fails during execution, the RECTIFIER module is used. It  predicts the reason for this failure from pixel input using vision language models and suggests a set of corrective subgoals.
>
> 3. We have added additional text descriptions of each of our modules in the paper. This text includes detailed descriptions, inputs/outputs, models used, and relations to other modules for the following aspects of HELPER: The `PLANNER`, `EXECUTOR`, `LOCATOR`, `RECTIFIER`, `INSPECTOR`, user feedback, and user personalization.
>
> Q: **The frequent use of “visuomotor” is a bit of an overclaim.**
>
> A: Thank you for pointing this out. We have removed all occurrences of this word in the paper, we agree that it can be misleading as we do not use any low-level skill visuomotor policies, given that the simulator API abstracts away from physics details.

---

### Official Review · Reviewer_hr3J · 2023-07-29

**Soundness:** 3

**Excitement:**

2: Mediocre: This paper makes marginal contributions (vs non-contemporaneous work), so I would rather not see it in the conference.

**Paper Topic And Main Contributions:**

This paper presented an embodied agent, HELPER, that grounds free-form human-robot dialogue instructions and corrections to visuo-motor programs. With the help of the memory-augmented pretrained LLM prompting method, the HELPER achieves a new SOTA on TEACh TfD benchmark.

**Questions For The Authors:**

1. How HELPER benefited from visual information? If the LLM Planner plans without vision input, how can it understand the current object status, such as if a bread is already sliced when the goal is to prepare a sandwich.

2. Could the HELPER be applied to other two tasks in TEACh dataset?

3. How many modules are there in total within the HELPER and how are they correlated? An independent figure that high-levelly shows the modules is highly encouraged to be added to the paper.

4. How the user feedback is collected? If the HELPER fails to succeed after getting the user feedback once, will it receive another user feedback?

**Reasons To Accept:**

The proposed HELPER uses a novel modular design including modules such as the Planner module, Rectifier module and etc. Extensive experiments are conducted to study the HELPER's performance. The personalization study shows the potential of the HELPER to be applied to more general tasks.

**Reasons To Reject:**

Here presents a few concerns based on the paper and more questions related to each concern are detailed in the next section.
1. The HELPER relies heavily on the text and it seems that the HELPER only uses image information when the execution fails and when searching for target objects.
2. The TEACh dataset also presents EDH task and TATC task, but HELPER is only evaluated in TfD task.
3. The modules in HELPER are not described clearly enough. Especially the input and output of each module lack some details.
4. User feedback misses some details.

**Reproducibility:**

2: Would be hard pressed to reproduce the results. The contribution depends on data that are simply not available outside the author's institution or consortium; not enough details are provided.

**Reviewer Confidence:**

4: Quite sure. I tried to check the important points carefully. It's unlikely, though conceivable, that I missed something that should affect my ratings.

**Typos Grammar Style And Presentation Improvements:**

Figure 2 contains too much information and blends too many key information together which makes reader lose focus easily. Break figure 2 into several figures would be beneficial.

---

> ### Author Rebuttal · Authors · 2023-08-27
>
> Q: **The TEACh dataset also presents EDH task and TATC task, but HELPER is only evaluated in TfD task. Could the HELPER be applied to other two tasks in TEACh dataset?**
>
> A: **Per your suggestion, we have evaluated HELPER on the EDH task of TEACH.** For EDH, we provide the LLM with the subgoals completed thus far by the Driver in addition to the input dialogue and instruct the LLM to only output the remaining program to complete the task. We use the agent history provided by the EDH episode to initialize our semantic map and object memory.  **HELPER achieves 18.3% success and 31.5% goal condition success on validation seen**. Previous SOTA in this task is Danli (Zhang et. Al. 2022)  with  16.89% success and 25.10% goal condition success.
> HELPER not only outperforms the baselines, but it does so without any in domain training or finetuning on the TEACh dataset. Our focus was initially on the TfD task because of its increased complexity and similar task structure to a robot deployed in a real household. We will include these results in the camera ready.
>
> The TATC task isn't yet available for comprehensive evaluation. Please refer to the TEACh github repository for more information. Nonetheless, our user feedback experiments did incorporate the "Progress Check" action from this task, showing HELPER's ability to handle task feedback. We will evaluate  HELPER on the TATC task upon its full release.
>
> Q: **The HELPER relies heavily on the text and it seems that the HELPER only uses image information when the execution fails and when searching for target objects.**
>
> A: Thank you for your comment.
>
> Our method in fact uses more visual information than the baselines! It uses visual information in the following ways:
> We detect objects in 3D and track their location over time in our 3D object memory map, which we use for object search.   Visual image information is also used for implementing the manipulation and navigation actions predicted by the LLM, e.g., manipulating an object requires segmenting the object in the input 2D image.  Our error correction relies on visual data and uses off-the-shelf Vision-Language Models (VLMs) for discerning causes of failures. Section 4.2 demonstrates the importance of this feature in our pipeline. **On the contrary, previous methods, (e.g., Danli; Zhang et. Al. 2022) often use manually-defined subgoal error correction without visual context.**
>
> You are right that the LLM does not plan with visual inputs, and that visually-grounded planning is helpful for these tasks. Instead, our model adjusts the LLM's plan on the fly based on object attributes  (open/close, clean/dirty, open/close), which are used in pre-condition checks. For instance, if the bread is already sliced, the task of slicing is bypassed. At submission time, we used a mechanism that initializes object attributes to default values and updates them when actions are executed (like slicing). Although this method meets the requirements of the Teach benchmark, we acknowledge that real-world scenarios need visual attribute classification, as you are pointing out!
>
> **Following up on your suggestion, we have implemented a vision-based attribute classification** for each object added to our object memory.      We replace the default attribute initialization with CLIP-based attributed  classification. Specifically, we obtain the boolean value of each attribute by taking the best match of the image crop around each detected object with text prompts such as “The {object label} is {attribute}” and “The {object} is not {attribute}”.
> **Using the  CLIP attribute classifier for object attribute initialization instead of the default one, we achieve similar performance (-0.4% success difference; -0.1% goal condition difference; valid unseen)**. This shows that the default initialization of object attributes we used in the paper is sufficient for the TfD task. However, we anticipate  a general method beyond dataset biases of TEACh would much benefit from such vision-based attribute classification, and we thank you for raising this point.
>
> Q: **Ambiguity in HELPER's module descriptions. Would be hard pressed to reproduce the results.**
>
> A: Thank you for your feedback. To improve reproducibility and ease of understanding, we have  added an accessible repository link containing our code in the revised version of the paper, as we are not able to include code here. All models and tools used are all publicly available, including OpenAI's API for LLM inference, and perception models like SOLQ, ZoeDepth, and ALIGN. It's worth noting that the perception modules or LLM are modular and can be swapped with alternative networks as desired.
>
> To improve clarity, we have added an algorithm box, a visualization to demonstrate the interaction of the modules, and the following text:
>
> > The HELPER system is an embodied agent designed to convert natural language instructions and RGB sensor data into a sequence of robot actions. The system has a modular organization outlined in Algorithm S1 and visualized in Figure S4. The PLANNER module uses nearest neighbor retrieval to obtain the most relevant task plans from a memory bank by matching language encoding of accompanying dialogue. The retrieved language-plan pairs are used as in-context examples inside the prompt for an LLM to transform user instructions or dialogue into an executable program. This generated program is then sent to the EXECUTOR module, which translates each program step into specific navigation and manipulation actions. To support this, the system uses RGB sensor data to produce segmentation and depth images at each time step. These images are used for object identification and for building a semantic map, which is subsequently used for object tracking and navigation. Before executing each subgoal, the INSPECTOR module verifies if the necessary preconditions for an action, such as the robot already holding an object, are met. If not, the plan is adjusted according to the current environmental and agent state. Should a subgoal involve an undetected object, the EXECUTOR calls on the LOCATOR module. This module utilizes previous user instructions and LLMs' commonsense knowledge to efficiently search for the required object. If any action fails during execution, the RECTIFIER module is used. It  predicts the reason for this failure from pixel input using vision language models and suggests a set of corrective subgoals.
>
> 2. We have added additional textu descriptions of each of our modules in the paper. This text includes detailed descriptions, inputs/outputs, models used, initializations, and relations to other modules for the following aspects of HELPER: The `PLANNER`, `EXECUTOR`, `LOCATOR`, `RECTIFIER`, `INSPECTOR`, user feedback, user personalization.
>
> Q: **User feedback misses some details.**
>
> A: We generate user feedback as explained in the following text. We have added this to the paper.
>
> > In the user feedback evaluation, once the agent has indicated completion of the task from the original input dialogue, the agent will query feedback from the user. If the simulator indicates success of the task, the agent will end the episode. If the simulator indicates the task is not successful, feedback will be given to the agent for additional planning. This feedback is programatically generated from the TEACh simulator metadata, which gives us information about if the task is successful, and what object state changes are missing in order to complete the task (e.g., bread slice is not toasted, etc.). For each object state that is incorrect, we form a sentence of the following form: "You failed to complete the subtask: {subtask}. For the object {object}: {description of desired object state}." We combine all subtask sentences to create the feedback. HELPER follows the same pipeline (including examples, retrieval, planning, etc.) to process the feedback as with the input dialogue in the normal TfD evaluation. We show experiments with one and two user feedback requests in Section 4.2 of the main paper (a second request is queried if the first user feedback fails to produce task success).
>
> Q: **Figure 2 contains too much information and blends too many key information together which makes reader lose focus easily. Break figure 2 into several figures would be beneficial.**
>
> A: Thank you for your feedback. Per your suggestion, we have broken down the figure into several figures, and included figures for each module in the revised supplemental.

---

### Official Review · Reviewer_xAvR · 2023-07-31

**Soundness:** 3

**Excitement:**

3: Ambivalent: It has merits (e.g., it reports state-of-the-art results, the idea is nice), but there are key weaknesses (e.g., it describes incremental work), and it can significantly benefit from another round of revision. However, I won't object to accepting it if my co-reviewers champion it.

**Missing References:**

The "context-dependant LLM prompting" is highly related to retrieval-augmentation. For example [1] retrieve the top-k image caption candidates from an external datasource to augment the generation of a caption for an image (similarly context-dependant LLM prompting retrieves programs to produce a program).

- [1] Sarto, S., Cornia, M., Baraldi, L., & Cucchiara, R. (2022, September). Retrieval-augmented transformer for image captioning. In Proceedings of the 19th International Conference on Content-based Multimedia Indexing (pp. 1-7).

The "memory expansion" is a variant of continual learning [2] that has been studied for example in [3] to map new natural language inputs to program execution and in [4] to improve an instruction following agent.

- [2] Timothée Lesort, Vincenzo Lomonaco, Andrei Stoian, Davide Maltoni, David Filliat, Natalia Díaz-Rodríguez:
Continual learning for robotics: Definition, framework, learning strategies, opportunities and challenges. Information Fusion
Volume 58, June 2020, Pages 52-68.

- [3] Sida I. Wang, Samuel Ginn, Percy Liang, and Christopher D. Manning. 2017. Naturalizing a Programming Language via Interactive Learning. In Proceedings of the 55th Annual Meeting of the Association for Computational Linguistics (Volume 1: Long Papers), pages 929–938, Vancouver, Canada. Association for Computational Linguistics.

- [4] Ta-Chung Chi, Minmin Shen, Mihail Eric, Seokhwan Kim, Dilek Hakkani-Tür: Just Ask: An Interactive Learning Framework for Vision and Language Navigation. AAAI 2020: 2459-2466


**Paper Topic And Main Contributions:**

This work presents a system of various pre-trained components (Planner, Rectifier, VLM, Executor, Locator, Inspector) that achieves a new state-of-the-art success rate on the TEACh-TfD benchmark. The length weighted success is only about the same or similar to its competitors (Episodic-Transformer, JARVIS, FILM and DANLI).

The authors built upon a technique that uses LLMs to produce executable code to navigate an agent in a virtual household environment and add a retrieval-augmentation memory to the mix. The memory mechanism is responsible to retrieve previously successfully executed programs given a natural language dialogue, so that these are placed into the the language model's prompt template as examples. This approach can also be used as a continual learning technique by to adding more programs after deployment. And the authors add a mechanism to the replan and recover plans.

**Questions For The Authors:**

- Question A: How often is a program from the examples copied or nearly copied to the output? Is the main load on the retrieval augmentation?
- Question B: How do you initialize the memory with dialogue-program pairs?
- Question C: How well performs HELPER without the Inspector? Why is it missing in the ablation? Is this method commonly used also in other work compared to?
- Question D: The overall success rate is still quite low. What are the failures cases? Do you have an estimate for human or a perfect-planner performance?

**Reasons To Accept:**

- a novel demonstration of connecting various pre-trained components into a working system
- almost no training involved (only fine-tuning the object and depth detector)
- new state-of-the-art in TEACh-TfD w.r.t. success rate and an ablation that shows the benefit of program retrieval (2.12%)

**Reasons To Reject:**

- the presentation lacks clarity on the overall interaction between the various components (Planner, Rectifier, VLM, Executor, Locator, Inspector) and the general execution flow
- the comparison with other techniques seems hard because of the bias that the Inspector component induces ("the inspector verifies the preconditions [...] are all met [...] and adjusts the plan accordingly (L386-394)")
- missing results for openly available (huggingface) models (only GPT-3 and GPT-4 were tested)

**Reproducibility:**

2: Would be hard pressed to reproduce the results. The contribution depends on data that are simply not available outside the author's institution or consortium; not enough details are provided.

**Reviewer Confidence:**

3: Pretty sure, but there's a chance I missed something. Although I have a good feel for this area in general, I did not carefully check the paper's details, e.g., the math, experimental design, or novelty.

**Typos Grammar Style And Presentation Improvements:**

- L013: "our model executes the program" -- how can a model execute a program? maybe rephrase?
- L072: semicolon should be replace by "and"
- L085: "PEsonalized" -- is missing an "r"
- L096: "prompts helps the agent to parse language inputs" -- I do not directly see the connection here: is the idea that text helps to parse text?
- L109: "HELPER navigates home environments" -- virtual (!) home environments
- L359: "it uses data" -- the data should be formally described or at least an example should be presented
- L514: missing reference to Table 2

General Notes:

- Figure 2: B is not showing what "happens" with the program; the connection to A should be made more clear
- the general flow should be depicted: as far as I understand there is a loop involved, the VLM checks at each step for problems, then the Rectifier gets involved and then the plan is proceeded; but how can the plan proceed when the state might have changed?

References:

- lots of pre-prints (>25!), arxiv references should be checked again e.g. Progprompt is actually published in ICRA 2023
- "2022. New and improved embedding model." is missing authors and venue
- "Multilingual vision-and-language navigation with dense spatiotemporal grounding" is missing authors and year

---

> ### Author Rebuttal · Authors · 2023-08-27
>
> Thank you for your thorough comments. We address your concerns below.
>
> Q: **How well does HELPER perform without the Inspector? Why is it missing in the ablation? Is this method commonly used also in other work compared to?**
>
> A: **We conducted the ablation study you suggested by removing the Inspector component and  observed only a 0.4% success rate decrease without the Inspector component.**  This shows the influence of our pre-conditions checks on HELPER's performance is minimal. We have since included this ablation in the revised manuscript.  We would like to clarify that, similar to HELPER, **the most competitive previous work, notably Danli (Zhang et. Al. 2022), incorporates pre-condition checks as part of their FormPDDLProblem component** and error correction module. This demonstrates that our approach is in line with the baselines, and thus directly comparable with them. Additionally, **our Inspector module does not use any  privileged information** that is not standard in the TfD TEACh evaluation. This ensures that our comparison with other techniques is fair.
>
> Q: **The length weighted success is only about the same or similar to its competitors.**
>
> A: The reason for this is the following:  Our depth estimation network is much more accurate for near depths than for far  depths, as discussed in Section 4.2. **This leads to an extended mapping time, affecting the length-weighted success.** **When employing ground truth depth, our approach not only maintains a significantly higher success rate but also exhibits a substantially improved path weighted metric compared to baselines.** Thus, when a precise depth sensor (e.g., LiDAR) is used or a more accurate depth estimation network, our method is expected to surpasses the baseline models in path-weighted metrics. We will add this explanation to the text.
>
> Q: **Missing results for openly available (huggingface) models (only GPT-3 and GPT-4 were tested)**
>
> A: The OpenAI Python API we use is publicly available and we use the GPT-4-0613 version. It is widely used on several paper publications. The reason is that models currently hosted on huggingface are weaker in code generation.
>
> Q: **The presentation lacks clarity on the overall interaction between the various components (Planner, Rectifier, VLM, Executor, Locator, Inspector). Would be hard pressed to reproduce the results.**
>
> A: Thank you for the constructive feedback regarding HELPER's module clarity.
> To address your concerns, we have added an algorithm box, a visualization to demonstrate the interaction of the modules, and the following text:
>
> > The HELPER system is an embodied agent designed to convert natural language instructions and RGB sensor data into a sequence of robot actions. The system has a modular organization outlined in Algorithm S1 and visualized in Figure S4. The PLANNER module uses nearest neighbor retrieval to obtain the most relevant task plans from a memory bank by matching language encoding of accompanying dialogue. The retrieved language-plan pairs are used as in-context examples inside the prompt for an LLM to transform user instructions or dialogue into an executable program. This generated program is then sent to the EXECUTOR module, which translates each program step into specific navigation and manipulation actions. To support this, the system uses RGB sensor data to produce segmentation and depth images at each time step. These images are used for object identification and for building a semantic map, which is subsequently used for object tracking and navigation. Before executing each subgoal, the INSPECTOR module verifies if the necessary preconditions for an action, such as the robot already holding an object, are met. If not, the plan is adjusted according to the current environmental and agent state. Should a subgoal involve an undetected object, the EXECUTOR calls on the LOCATOR module. This module utilizes previous user instructions and LLMs' commonsense knowledge to efficiently search for the required object. If any action fails during execution, the RECTIFIER module is used. It predicts the reason for this failure from pixel input using vision language models and suggests a set of corrective subgoals.
>
> We have added additional text descriptions of each of our modules in the paper. This text includes detailed descriptions, inputs/outputs, models used, and relations to other modules for the following aspects of HELPER: The `PLANNER`, `EXECUTOR`, `LOCATOR`, `RECTIFIER`, `INSPECTOR`, user feedback, user personalization.
>
> To improve reproducibility and ease of understanding, we have also added an accessible repository link containing our code in the revised version of the paper, as we are not able to include code here.
>
> All models and tools used are all publicly available, including OpenAI's API for LLM inference, and perception models like SOLQ, ZoeDepth, and ALIGN. It's worth noting that the perception modules or LLM are modular and can be swapped with alternative networks as desired.
>
> Q: **How often is a program from the examples copied or nearly copied to the output? Is the main load on the retrieval augmentation?**
>
> A: Our memory bank contains only 19 examples, which are derived from training samples. These examples do not span the variety of dialogue and tasks in the evaluation set. Consequently, the LLM cannot succeed by merely replicating  these examples, instead it uses them as in-context references. We did not train the retriever, but instead used  pre-trained language encoders.  Even without retrieval (Section 4.2), the LLM's performance is consistent with the previous SOTA. This showcases the LLM's capability to generate accurate outputs, even in the absence of the most relevant examples, showcasing its potential for more than simply copying.
>
> Q: **How do you initialize the memory with dialogue-program pairs?**
>
> A: We start with a memory bank of 19 examples (11 for the planner and 8 for the rectifier). These examples were selected from the first 100 training samples. Each dialogue from the chosen examples is paired with a program, manually written by a domain expert familiar with the task.
>
> Our approach required merely a few hours to initialize the memory bank. Despite its simplicity and speed, our model outperforms previous methods that necessitated months of data collection and training. For instance, the TEACh author's process of collecting 1482 training demonstrations and subsequent data cleaning is considerably more time-consuming. Additionally, the top-3 competitive baselines finetune an LLM to predict subgoals, which also demanded extended durations.  Despite being more efficient, our method displays an increased generalization to diverse input text, like user feedback and user customization requests (Sections 4.2 & 4.3), even when such feedback considerably differs from the dialogues in the training set.
>
> Q: **What are the remaining failure cases?**
>
> A: As you are pointing out, the task remains challenging, and there's room for improvement. The evaluation includes only episodes where human performers succeeded.
>
> The main failure cases are as follows:
>
> - **Perception failures**
> We observe a 2X increase in success rate using ground truth 3D object segmentation and labelling. Our system's modularity allows us to  replace our object segmentator and labeller with better ones  as they become available.
>
>
> - **Task Complexity** As tasks become more complex and long time horizon that require additional subgoals, we noticed a decrease in success rate. This decline is attributed to errors propagating across subgoals, sometimes arising from the LLM planner's inaccuracies, occasional error correction failure of a subgoal, or ineffective retrieval of task-relevant examples.
>
> - **Object Referential Grounding** Our current model has limitations when pinpointing objects described in referential noun phrases, like "second cabinet to the left of the sink." If a general category-based search does not find the object, our search module might fail in locating the target object.
>
>
>
> Q: **Missing references for retrieval-augmentation and continual learning**
>
> A: We have added all the references you are pointing out to our related work with appropriate descriptions.
>
> Q: **References and typos**
>
> A: We sincerely appreciate your attention to detail. All the typos and references issues you pointed out have been corrected in the revised manuscript.
>
> Q: **How can the plan proceed when the state might have changed by the rectifier?**
>
> A: Thank you for bringing up this point of clarification. If the state changes, this does not affect the plan execution process. We do maintain an object memory map that keeps track of each object and its visual attributes (e.g., sliced, clean). The plan is adjusted on the fly based on these object attributes  (open/close, clean/dirty, open/close), which are critical during our *pre-condition checks*. The system evaluates the LLM's current plan based on the present object attributes and  dynamically adjusts it. For instance, if the bread is already sliced, the task of slicing is bypassed. At submission time, we used a mechanism that initializes object attributes to default values and updates them when actions are executed (like slicing). We will clarify this in the paper as part of the module and figure revisions.

---

### Meta-Review · Area_Chair_6axt · 2023-09-19

**Recommendation:** 4

**Metareview:**

This paper proposed a new system containing mostly pre-trained components for the TEACH benchmark of virtual mobile manipulation from dialog history or instructions.

The strengths highlighted by reviewers are:
- Good relevance to robotics
- Strong results of the proposed method
- Convincing empirical evaluation on standard benchmarks and baselines (this was initially lacking, but got significantly improved during rebuttal with additional experiments)
- Method requires minimal training.

Outstanding weaknesses after rebuttal are:
- Organization of the paper and confusion over how the many modules interact. The authors have attempted to address it.

---

### Decision · Program_Chairs · 2023-10-07

**Decision:**

Accept-Findings

**Comment:**

This paper proposed a new system containing mostly pre-trained components for the TEACH benchmark of virtual mobile manipulation from dialog history or instructions.

The strengths highlighted by reviewers are:
- Good relevance to robotics
- Strong results of the proposed method
- Convincing empirical evaluation on standard benchmarks and baselines (this was initially lacking, but got significantly improved during rebuttal with additional experiments)
- Method requires minimal training.

Outstanding weaknesses after rebuttal are:
- Organization of the paper and confusion over how the many modules interact. The authors have attempted to address it.